

# DASH: A MATLAB Toolbox for Paleoclimate Data Assimilation

Jonathan King[1,2], Jessica Tierney[1], Matthew Osman[1,3], Emily J. Judd[4], and Kevin J. Anchukaitis[1,2,5]

[1]Department of Geosciences, University of Arizona, Tucson, Arizona, USA
[2]Laboratory of Tree Ring Research, University of Arizona, Tucson, Arizona, USA
[3]Department of Geography, University of Cambridge, Cambridge, UK
[4]Department of Paleobiology, Smithsonian National Museum of Natural History, Washington, DC, USA
[5]School of Geography, Development, and Environment, University of Arizona, Tucson, Arizona, USA

**Correspondence:** Jonathan King (jonking93@email.arizona.edu)

**Abstract.** Paleoclimate data assimilation (DA) is a novel tool for reconstructing past climates that directly integrates proxy records with climate model output. Despite the potential for DA to expand the scope of quantitative paleoclimatology, these methods remain difficult to implement in practice due to the multi-faceted requirements and data handling necessary for DA reconstructions, the diversity of DA methods, and the need for computationally efficient algorithms. Here, we present DASH, a

MATLAB toolbox designed to facilitate paleoclimate DA analyses. DASH provides command line and scripting tools that implement common tasks in DA workflows. The toolbox is highly modular and is not built around any specific analysis, and thus DASH supports paleoclimate DA for a wide variety of time periods, spatial regions, proxy networks, and algorithms. DASH includes tools for integrating and cataloguing data stored in disparate formats, building state vector ensembles, and running proxy (system) forward models. The toolbox also provides optimized algorithms for implementing ensemble Kalman filters,

particle filters, and optimal sensor analyses with variable and modular parameters. This paper reviews the key components of the DASH toolbox and presents examples illustrating DASH's use for paleoclimate DA applications.

## 1 Introduction

Past climates provide insight into the drivers, variability, and evolution of the Earth's climate system, and are invaluable for providing insight on the consequences of current and future anthropogenic climate change (Alley, 2003; Hargreaves et al.,

2007; Rice et al., 2009; Schmidt, 2010; Snyder, 2010; Ault et al., 2014; Coats et al., 2020; Tierney et al., 2020a). Paleoclimate studies can help constrain important climate system properties including Equilibrium Climate Sensitivity (Hegerl et al., 2006; Rohling et al., 2012; Hansen et al., 2013; Kutzbach et al., 2013; Rohling et al., 2018; Sherwood et al., 2020; Tierney et al., 2020b; Zhu et al., 2020b), quantify internal and forced variability across a range of timescales and climate system metrics (Cane et al., 2006; Cook et al., 2011; Goosse et al., 2012a; Ault et al., 2013; Fernández-Donado et al., 2013; Anchukaitis et al.,

2019; Neukom et al., 2019; Fang et al., 2021), and can serve as analogues for future warm climate states projected to occur due to anthropogenic warming (Overpeck et al., 2006; Burke et al., 2018; Tierney et al., 2020a, 2022). Reconstructions of past climates also provide out-of-sample targets used to assess the skill of climate models, which in turn helps constrain future





projections and enables superior climate change adaptation strategies (Crowley, 1991; Hargreaves and Annan, 2009; Schmidt et al., 2013; Zhu et al., 2021a, b; Gulev et al., 2021).

Beyond the limited period of instrumental climate observations, researchers have primarily relied on two methods for studying past climates: proxy reconstructions and climate model hindcasts. In a proxy reconstruction, paleoclimatologists use climate proxy records, such as tree rings, ice cores, speleothems, corals, and lake and marine sediments, to make statistical estimates of past climates. These reconstructions rely on a combination of empirical and process-based understanding to link proxy records to features and characteristics of the Earth's climate system. A major advantage of using proxy data to reconstruct past cli-
mates is that they produce estimates of temperature, precipitation, or other climate variables that are consistent with the actual trajectory of the Earth's climate system. These reconstructions can also provide independent validation of climate model performance. However, many factors can hinder the inference of past climates from proxy data. These factors include the sparse distribution of proxy records through space and time, time-uncertainty due to limits on the precision of geochronology, and the influence of multivariate or non-climatic factors on proxy records. Furthermore, the physical processes that archive climate
signals in proxy records can be complex and are often not completely understood, which complicates the extraction of climate signals from proxy data using linear, univariate, and empirical statistical approaches. Proxy records are sensitive to the local climates in which they form, but many reconstructions target large-scale climate features or ocean-atmosphere modes not directly sensed by the available proxy data. Some reconstructions derive relationships between proxy records and target variables using calibrations with the instrumental era; however, modern climate is not in equilibrium and continues to respond to increasing
anthropogenic climate forcings. Therefore, modern teleconnections and climate-system spatial covariance patterns may differ from long-term and unforced patterns. Finally, many proxy reconstruction methods assume that teleconnections between local- and large-scale climate variables are stationary over reconstruction periods, an assumption that may not hold in reality.

    Climate model hindcasts leverage general circulation models to simulate past climate states using estimates of past boundary conditions, such as the Earth's orbital parameters, atmospheric greenhouse gas concentrations, volcanic eruptions, continental
configurations, and land cover. By contrast with proxy reconstructions, climate model hindcasts simulate data for target climate variables at all spatial points and time steps within the model domain. Furthermore, these simulated climate variables evolve according to fundamental physical governing equations and parameterizations, rather than the statistical associations and assumptions typically used for proxy reconstructions. Consequently, paleoclimate simulations can provide insight into the physical mechanisms behind reconstructed climate phenomena. However, no model fully captures the real Earth system, and
determining appropriate boundary conditions becomes increasingly difficult going back through geologic time, so all paleoclimate hindcasts necessarily contain errors in their representation of past climates. Additionally, many model variables are dominated by internal variability, sensitivity to initial conditions, and/or chaotic behavior over a range of time periods (Deser et al., 2012). Thus, no individual simulation will capture the true or specific trajectory of the Earth's past climate; instead, each simulation represents a single possible trajectory in a distribution of physically-plausible past climate states (e.g. Kay et al.,
2015). Finally, climate models require external validation to evaluate their fidelity and accuracy in reproducing past climate states.





Recently, data assimilation (DA) methods have emerged as a novel approach to the problems and challenges of paleoclimate reconstruction (e.g. Bhend et al., 2012; Goosse et al., 2012b; Mairesse et al., 2013; Hakim et al., 2016; Steiger et al., 2017, 2018; Tardif et al., 2019; Tierney et al., 2020b; King et al., 2021; Osman et al., 2021; Zhu et al., 2022; King et al., 2022). Unlike

the two independent approaches described above, DA methods integrate proxy data directly with climate model output and thereby leverage the strengths of both information sources. Because they utilize climate model simulations, DA methods can provide full-field global reconstructions (e.g. Evans et al., 2001) for nearly any simulated climate parameter, since the relationships between variables are linked through the physically based governing equations of the model. Simultaneously, DA reconstructions are constrained by proxy records and thus reflect the true trajectory of the Earth's past climate. DA methods use

forward models to describe how climate signals are sensed by and recorded in proxy archives, and thus can incorporate proxy system physical processes that are multivariate or nonlinear. Furthermore, the use of proxy forward models allows DA methods to relax calibration requirements when attempting to reconstruct large-scale climate modes or fields, such that proxy data can be calibrated to local climate variables rather than directly to large-scale teleconnections. DA methods also relax assumptions of teleconnection stationarity, as the effects of changing climate boundary conditions can be reflected in the evolution of climate

model output and its covariance.

Despite the potential strengths of paleoclimate DA, these reconstructions are often difficult to implement in practice. DA analyses require numerous discrete tasks, including preparing and integrating the output from climate model simulations, proxy records, and possibly instrumental data, all of which may use different data formats, units, and metadata. The number of potential reconstruction targets and proxy variables is immense, and the choice of algorithm parameters will affect the

implementation of any particular DA reconstruction (compare Tardif et al., 2019; Tierney et al., 2020b; King et al., 2021; Osman et al., 2021; King et al., 2022). Consequently, it can be difficult to adapt codes implementing an existing reconstruction to alternative applications. Paleoclimate DA also encompasses a diverse array of algorithms and algorithm variants (compare Goosse et al., 2006; Dubinkina and Goosse, 2013; Steiger et al., 2014; Matsikaris et al., 2015; Comboul et al., 2015; Dee et al., 2016; Acevedo et al., 2017; Liu et al., 2017; Perkins and Hakim, 2017; Franke et al., 2020), further increasing the complexity

of implementing DA codes. Finally, DA methods are often computationally intensive and require both optimized algorithms and efficient use of computer memory, and these considerations can dissuade potential users lacking experience or access to high-performance computing.

Although DA software does exist, thus far these packages are not suitable for generalized paleoclimate applications with a diverse range of time scales, climate model requirements, and proxy data types. Packages designed to implement general DA

methods typically lack support for fundamental components of paleoclimate DA, such as the use of proxy forward models. By contrast, DA packages designed for paleoclimate applications, such as the LMR (Hakim et al., 2016; Tardif et al., 2019) or PHYDA (Steiger et al., 2018), have been built to implement specific analyses, use particular proxy data, or incorporate specified climate model inputs. Adapting these products for generalized paleoclimate applications requires modifying the source code, which may be difficult or time-intensive and thus presents a barrier to their use.

A second difficulty for paleoclimatologists seeking to implement DA is that the methods are comparatively complex relative to existing reconstruction methods. Describing experimental DA setups in sufficient detail to allow reproducibility requires



considerable length, and published methods may focus of the broad scope of the mathematics while neglecting the details of key implementation steps in favor of brevity. Additionally, there are still relatively few paleoclimate applications in the mathematical DA literature, so DA descriptions may use a variety of mathematical notations. Finally, the diversity of algorithm

variants potentially hinders transparency and accessibility, as studies using similarly named algorithms may implement different methods in practice (compare Tardif et al., 2019; Franke et al., 2020; Tierney et al., 2020b; King et al., 2022). Ultimately, there are limited frameworks for discussing DA within the paleoclimate literature, and the field as a whole would benefit from more transparent implementations that do not require additional specialized training.

    In this paper, we present DASH, a MATLAB toolbox supporting paleoclimate data assimilation. The toolbox is designed for

general paleoclimate DA and is not built around or for any particular analysis, time period, proxy type, or climate model. Consequently, the toolbox is highly modular and allows flexible implementation of diverse DA analyses. DASH provides command line / scripting utilities designed to implement common tasks for paleoclimate DA workflows, with a goal of improving access to DA methods for users with diverse scientific backgrounds. DASH includes support for organizing climate data, building state vector ensembles, running proxy forward models, and implementing standard DA algorithms. All algorithms are optimized for

both speed and efficient memory use. Our goal is for DASH to improve clarity and transparency in DA analyses and provide a framework for paleoclimate DA discussions. Consequently, DASH commands are designed to provide a description of their routines, thereby promoting the creation of human-readable DA scripts.

    The remainder of this paper is organized as follows. In Section 2, we present a brief overview of paleoclimate DA, with the aim of introducing common tasks, data, and algorithms for paleoclimate DA workflows. In Section 3, we describe the

DASH toolbox specifically. We detail its general characteristics and layout and highlight its major components. In Section 4, we provide two examples that use DASH to implement paleoclimate data assimilation. These examples use different temporal periods, spatial regions, proxy networks, and algorithms in order to demonstrate the flexibility of the DASH toolbox. Finally,in Section 5 we provide a set of best practices and caveats, discuss the DASH toolbox in the broader context of paleoclimate DA, and outline potential and anticipated future developments to the code.

## 115   2   Overview of Paleoclimate DA

This section provides a brief overview of paleoclimate data assimilation, with the goal of introducing DA to paleoclimate researchers who may not be familiar with the broader mathematical DA literature. In particular, we aim to (1) Provide accessible insight into the DA "black-box", (2) improve the transparency of common DA algorithms, (3) establish a vocabulary for DA workflows, and (4) provide context for the DASH software package. We focus on illustrating the tasks and quantitative routines

most frequently used in paleoclimate DA workflows, rather than providing complete mathematical descriptions (which can be found elsewhere, e.g. Evensen (1994); Van Leeuwen (2009)). Here, we focus specifically on the ensemble Kalman filter and ensemble particle filter methods. We also describe an optimal sensor algorithm based on an ensemble Kalman filter framework. Additional and more complete descriptions of DA algorithms are available in Evensen (1994); Anderson and Anderson (1999); Whitaker and Hamill (2002); Goosse et al. (2006, 2012b); Dubinkina and Goosse (2013); Steiger et al. (2014); Comboul et al.



(2015); Hakim et al. (2016); Tardif et al. (2019); Franke et al. (2020); Tierney et al. (2020b); King et al. (2021); Osman et al. (2021).

## 2.1 Conceptual framework

In the broadest terms, DA methods combine output from climate model simulations ($X_p$) with proxy records (Y) to reconstruct a set of climate variables ($X_a$).

$$X_a = f(X_p, Y) \tag{1}$$

The reconstructed climate variables $X_a$, also known as the analysis, are calculated by updating climate variables from the climate models $X_p$ to more closely match the proxy records Y. The Kalman filter and particle filter methods discussed in this paper can also be formulated as Bayesian filters (Chen et al., 2003; Wikle and Berliner, 2007) wherein new information (Y) is used to update estimates of state parameters (X). Hence, we will often refer to $X_p$ and $X_a$ as the *prior* and *posterior*,

respectively.

In general, climate model output is organized into *state vectors*, which consist of multi-dimensional spatiotemporal climate model output reshaped into a vector of data values (Figure 1, upper left). There is no strict definition for the contents of a state vector, but they typically include data for one or more climate variables at a set of spatial points. A state vector might also contain a trajectory of successive points in time; for example, individual months of the year or a number of successive years

following an event of interest. Essentially, a state vector serves as a possible description of the climate system for some period of time. In this paper, we focus on *ensemble* DA methods, which rely on state vector ensembles. A *state vector ensemble* is a collection of multiple state vectors organized in a matrix (Figure 1, upper right), and a given ensemble provides an empirical distribution of possible climate states. For paleoclimate applications, ensemble members are often selected from different points in time, different members of an initial condition, perturbed physics, or multi-model ensemble, or a combination of

these options. In a typical DA algorithm, the state vectors in an ensemble are compared to a set of proxy record values for a given time step or time slice. Essentially, the method compares the potential descriptions of the climate system taken from the climate model to the proxy values from the real past climate record. The similarity of each state vector to the set of proxy records is then used to inform the final reconstruction.

In order to compare state vectors with a set of proxy record values, DA methods must transfer state vectors and proxy records

into a common unit space. This is accomplished by applying proxy forward models (Evans et al., 2013) to relevant climate variables stored in each state vector (Figure 1, bottom left). For a given state vector, a forward model is run for each proxy record in $\hat{Y}$, using the relevant climate variables in the state vector as inputs. This produces a set of values in the same units as the proxy records and therefore allows direct comparison of the state vector and observed proxy values. In general, DA methods will run a forward model to estimate each proxy record for each state vector in an ensemble; the collective outputs

are referred to as *proxy estimates* ($\hat{Y}$) and allow comparison of the states in the ensemble with a set of proxy records. The difference between the proxy observations and proxy estimates is known as the *innovation* (Figure 1, bottom right):

$$\text{innovation} = Y - \hat{Y} \tag{2}$$



and describes the discrepancies between the actual proxy records and the climate states in the ensemble. The innovation is then used to constrain or update the prior ensemble ($X_p$) to more closely resemble the observed proxy records.

In addition to proxy innovations, the DA methods detailed here also consider proxy uncertainties ($R$) when comparing state vectors to the proxy records, such that:

$$X_a = f(X_p, Y, R) \tag{3}$$

In this way, proxy records with high uncertainties are given less weight in the reconstruction. In classical assimilation frameworks, $R$ is often derived from the uncertainty inherent in measuring an observed quantity. For example, $R$ might reflect the uncertainty of width measurements in a tree-ring chronology. However, in nearly all paleoclimate applications, measurement uncertainties are small compared to (1) the uncertainties inherent in proxy forward models, and (2) uncertainties resulting from non-climate signals (i.e. noise) archived in the proxy records. Thus, in paleoclimate DA the proxy uncertainties $R$ must account for proxy noise and forward model errors, as well as the covariance between different proxy uncertainties. Most generally, $R$ is the proxy error-covariance matrix. This matrix is diagonal when proxy errors are assumed uncorrelated; otherwise, $R$ is a full covariance matrix.

## 2.2 Update equations and algorithms

There are several different algorithms that can be used to pass the new information produced by the innovation into the climate models. One of the most commonly used methods in paleoclimate applications is the Kalman Filter (Kalman, 1960; Andrews, 1968; Evensen, 1994), and its update equation is given by:

$$X_a = X_p + K(Y - \hat{Y}) \tag{4}$$

Equation 4 shows that the innovation is weighted by the Kalman Gain matrix ($K$) in order to compute an update for each state vector in the prior ensemble ($X_p$). The Kalman Gain weighting considers multiple factors, including (1) the covariance of the proxy estimates ($\hat{Y}$) with target climate variables ($X_p$), (2) the covariance between the proxy estimates ($\hat{Y}$), and (3) the uncertainties in the proxies ($R$), such that:

$$K = \text{cov}(X, \hat{Y})[\text{cov}(\hat{Y}) + R]^{-1} \tag{5}$$

Applying the updates produces an updated (posterior) ensemble ($X_a$) with climate states that more closely resemble those recorded by the real proxy records ($Y$). The ensemble nature of $X_a$ is advantageous because the distribution of climate variables across $X_a$ can help quantify the uncertainty in the reconstruction.

By contrast with Kalman filters, particle filters (Van Leeuwen, 2009) combine the innovation with proxy record uncertainties ($R$)) to compute a weight for each state vector in the ensemble. The reconstruction is then calculated as a weighted mean of the state vectors in the ensemble. Classical particle filters compute these weights using a Bayesian scheme, such that each state vector $i$ is first assigned an importance weight:

$$s_i = \exp[-\frac{1}{2}(Y - \hat{Y}_i)^T \; R \; (Y - \hat{Y}_i)] \tag{6}$$



and then importance weights are normalized to give the final state vector weights:

$$w_i = \frac{s_i}{\Sigma_{j=1}^{N} s_j} \tag{7}$$

However, classical particle filters can suffer from degeneracy in the high-dimensional systems common to paleoclimate DA. Essentially, a single ensemble member receives a weight of 1, whereas all other ensemble members receive near-zero weights. When this occurs, reconstructed values ($X_a$) resemble the single state vector most similar to the proxy records, rather than values across an ensemble. A common correction for degeneracy involves using the mean of the $N$ state vectors with the

195 highest Bayesian weights. Alternatively, the "degenerate particle filter" refers to the case when the single best state vector is used as the reconstruction (e.g. Goosse et al., 2006, 2010). The "analogue method" may also refer to a degenerate particle filter (e.g. Goosse et al., 2006), although the meaning of this term varies throughout the paleoclimate literature.

When discussing Kalman and particle filters, it is important to distinguish between offline and online regimes. In an online regime, the assimilation updates are used to inform the evolution of the climate model simulations. Essentially, the updated

ensemble for a given time step informs the starting states of the climate model simulations in the next time step. By contrast, in the offline regime all climate model output is generated in advance, and so the assimilation updates do not inform the evolution of the climate model simulations (Oke et al., 2002; Evensen, 2003). Offline methods incur a significantly lower computational cost than online methods, but the priors of the reconstructed time steps are not constrained by the proxy records (Oke et al., 2002; Evensen, 2003; Matsikaris et al., 2015; Acevedo et al., 2017). As such, researchers must consider both computational

feasibility and the propagation of proxy information when choosing between the online and offline modes.

The optimal sensor algorithm described in this paper follows the method presented by Comboul et al. (2015). This method is derived from an ensemble Kalman filter and complements the reconstruction framework by providing additional information about the contribution of proxy data sites to the reconstruction. In paleoclimate, optimal sensor analyses have traditionally been used to evaluate the potential of new proxy sites, to prioritize future proxy development, and to assess the proxy network (i.e. the

210 collection of proxy records) necessary to skillfully reconstruct a climate field (e.g. Bradley, 1996; Evans et al., 1998; Comboul et al., 2015). Here, we expand the method to assess the relative influence of individual proxy records on a reconstructed index. Rather than reconstructing climate variables over time, the algorithm instead tests the ability of a proxy record to reduce the variance of a scalar climate metric J across an ensemble. A proxy record's ability to reduce variance is determined using the covariance of its estimates ($\hat{y}$) with the climate metric (J) combined with the uncertainty of the proxy record (R). For a given

proxy record (y), this equation is given by:

$$\Delta \sigma_k = \text{cov}(\hat{y}, J)^2 [\text{var}(\hat{y}) + R]^{-1} \tag{8}$$

and the proxy that most strongly reduces variance is selected as the optimal sensor:

$$s_{\text{optimal}} = \text{argmax} \ \Delta\sigma \tag{9}$$

This proxy is used to update the climate metric using an ensemble Kalman Filter (Equations 4, 5) and then removed from the

220 network. This analysis then iterates using the remaining sensors until the desired number of sensors are selected. Ultimately,




the method both ranks the proxies in a network and also assesses the total variance reduced by a particular proxy network. This method requires proxy estimates ($\hat{Y}$) to calculate climate metric covariance but does not use proxy record values themselves ($Y$), as the potential to reduce ensemble variance is independent of actual proxy values.

## 3    Description of DASH

### 225    3.1    General Characteristics

DASH is a MATLAB toolbox designed to help implement paleoclimate data assimilation. The code is designed for use from the command line as well as within scripts and functions. DASH is written in an object-oriented style, which supports the modularity of the code; the toolbox consists of several classes and packages, each implementing a common task for paleoclimate DA. The code is intended for users with basic previous experience with MATLAB; in particular, users will benefit from 230    knowing how to write a basic `for` loop, and how to index into arrays.

A stated goal of the DASH toolbox is to support the transparency of paleoclimate data assimilation analyses, and the object-oriented design supports this aim. DASH methods are accessed via dot-indexing, which improves clarity by placing sub-tasks within the context of a larger piece of the data assimilation process. Additionally, tasks with many parameters or options are organized into objects, which can store settings between commands. Consequently, the parameters used to implement complex 235    algorithms are split across several commands, improving both the clarity and modularity of codes utilizing DASH.

To support command-line workflows, DASH is designed for console display and does not rely on a graphical user interface (GUI). Users can inspect the state of class objects, assimilation analyses, and other DASH components by displaying them in the console. Users can also examine reference guides for DASH components using the `help` command; however, we recommend that users instead use the HTML documentation set, which is detailed below. Further, we are cognizant that users 240    may not be familiar with all aspects of paleoclimate data assimilation, or with all components of the toolbox. DASH therefore implements robust input checking and error handling for all user-facing methods. Error messages are designed to clearly communicate input failures and suggest possible solutions without requiring users to know the inner workings of the DASH codebase.

The DASH toolbox is accompanied by comprehensive documentation written in HTML. This documentation includes (1) a 245    reference guide for every class, package, method, and function, (2) tutorials for nearly all user-facing commands, and (3) How-Tos and FAQs for common tasks and troubleshooting. The entire documentation can be accessed by entering the `dash.doc` command from the MATLAB command line. Alternatively, users can open the reference manual for a particular component by providing the component name as input: » `dash.doc("component name")`. The documentation is also available on the project's website (https://jonking93.github.io/DASH).

To install DASH, users should first download a stable release of the toolbox, which can be found at the project's Github repository (https://github.com/JonKing93/DASH/releases), MATLAB FileExchange (https://www.mathworks.com/matlabcentral/fileexchange/120453-dash), or in the MATLAB Add-On Explorer. Then, open the downloaded `DASH-<version>.mltbx`





file to complete the installation. We encourage users to download one of the project's stable releases, as the source code on the Github repository's main branch may be in active development and is therefore not configured for quick installation.

## 3.2 DASH Components

DASH consists of several classes and packages, each implementing a particular task commonly required for paleoclimate data assimilation (Figure 2). In brief, the toolbox contains components to (1) organize and catalogue input data, (2) design and build state vector ensembles, (3) estimate proxy records via proxy forward models, and (4) implement common data assimilation algorithms. In the remainder of this section, we examine the characteristics and features of each of these modules. We realize that many aspects of these classes are abstract in concept, and therefore provide step-by-step tutorials in the DASH documentation that illustrate how DASH works in practice. The examples in Section 4 also demonstrate the use of common DASH commands, albeit in a less detailed style than the tutorials.

### 3.2.1 Organize Climate Data: `gridfile`, `gridMetadata`

We begin our overview with the `gridfile` class. This module facilitates the combination of datasets stored in different formats and with disparate metadata by creating data catalogues. The data catalogued within a `gridfile` is associated with user-specified metadata, which allows users to manipulate large datasets using preferred and human-readable metadata formats. This class thereby allows users to consolidate datasets split across multiple files, promotes human-readable data manipulation, and unites disparate data formats within an intuitive framework. The class implements `gridfile` objects, and each object implements a catalogue for data stored in various source files. The basis of each catalogue is an abstract $N$-dimensional grid, whose scope is defined by user-provided dimensional metadata. This allows users to catalogue datasets of varying dimensionality, while simultaneously tagging data elements with unique and user-preferred metadata values. We note that the grid abstraction does not imply that `gridfile` datasets must use a Cartesian spatial grid. Rather, the class supports a wide-variety of spatial layouts, including rectilinear systems, tripolar grids, randomly distributed spatial sites, and datasets without any spatial component at all.

After first defining the scope of a `gridfile`, users can add data source files to the catalogue by associating the data in each file with a portion of the $N$-dimensional grid. In this way, the data in each source file is placed within the context of the overall dataset. The `gridfile` package supports data source file formats common in paleoclimate DA – including NetCDF, OPeNDAP, MATLAB's binary MAT files, and delimited-text files – and individual catalogues may contain any mixture of file formats. The contents of each catalogue are saved in a `.grid` file, so data catalogues can persist across multiple coding sessions. We emphasize that these `.grid` files save only a *catalogue* of a dataset, and not the dataset itself. Thus, `.grid` files do not duplicate data, and individual `.grid` files remain small (typically a few kilobytes) even when they refer to datasets spanning many gigabytes of memory. Once a catalogue is complete, users can return data using the `load` command, which provides a common interface for accessing data in the catalogue. Users can also return a subset of the catalogued data by querying the associated metadata. The `gridfile` class also allows users to apply data transformations, such as log transforms or fill values, to a catalogue. Such transformations are only applied to loaded data, which improves computational efficiency and





maintains the data sources as read-only files. Finally, the class allows users to perform arithmetic operations like addition and multiplication across multiple gridfile datasets; these operations are analogous to several commonly used NetCDF operators but are not limited to NetCDF files.

The `gridfile` class relies on `gridMetadata`, which implements objects that define the metadata for a dataset. The `gridMetadata` class plays an auxiliary role within the DASH toolbox, and is mainly used to define the scope of `gridfile` catalogues and to locate data subsets within a `gridfile` dataset. We contrast `gridMetadata` with `ensembleMetadata`, a second metadata class implemented by DASH. Whereas `gridMetadata` characterizes values in an $N$-dimensional dataset, `ensembleMetadata` instead characterizes $N$-dimensional datasets after they are reshaped into state vector ensembles. Further details for the `ensembleMetadata` class are given in Section 3.2.3.

### 3.2.2 Build state vector ensembles: `stateVector, ensemble`

The next key component of DASH is the `stateVector` class. This component is designed to facilitate flexible design of state vector ensembles while minimizing the amount of data manipulation done by the user. The class implements objects that hold design parameters required to build a state vector ensemble from `gridfile` catalogues. To design a state vector, users first initialize a `stateVector` object and the climate variables that it will contain. Each variable is associated with a `gridfile` dataset, and multiple variables in the state vector may be derived from the same dataset (e.g., climate variables representing mean annual and mean summer temperatures could be drawn from the same monthly temperature catalogue). We note that when a user adds a variable to a `stateVector` object, no data is loaded into memory at that time. Instead, the object initializes a set of design parameters that can later be used to extract data for the variable from its `gridfile`. To design the state vector, users next specify options for the dimensions of the variables. As a first step, users should indicate which dataset dimensions are used to select ensemble members. In most paleoclimate DA applications, ensemble members are selected from different time steps and/or different climate model simulations. However, `stateVector` is highly flexible and also allows ensembles built along other dimensions; for example, ensembles built from different height levels or from different spatial locations and sites. Users can also specify a subset of elements along an ensemble dimension to use for building ensemble members. For example, in a dataset with monthly resolution, a user could specify to only select ensemble members from January time steps. The class also includes many additional methods for designing state vector variables: users can specify that a variable should be drawn from a subset of a `gridfile` dataset, or that it should be a computed mean, weighted mean, or sum total over some data dimension. Users can also select options for processing variables with different metadata formats, as well as specify that individual ensemble members should contain temporal sequences. For example, a variable could include data from individual months of the year, useful for seasonal analyses. Alternatively, a variable could hold values from successive years, which supports superposed-epoch analyses for climate conditions following discrete events of interest.

Once a design is complete, users can build the state vector ensemble using the `build` command. This command loads necessary data from the `gridfile` catalogues and builds a state vector ensemble according to the specified design parameters. When building a state vector ensemble, the `stateVector` class will ensure that all variables within a given ensemble member align to the same metadata values. For example, in an ensemble selected from different time steps, the data for the variables



in each ensemble member will all correspond to the same time step. Similarly, in an ensemble selected from different model simulations, the variables in each ensemble member will all be drawn from the same simulation. The class also ensures that ensemble members are constructed from complete data. For example, if a state vector variable includes a temporal mean or sequence, then the build method will never select an ensemble member for which the mean or sequence would extend outside of the bounds of the dataset.

When building an ensemble, users have the option to return the ensemble directly as an array, or to save the ensemble to a file. This later option is useful, as state vector ensembles may exceed the size of active memory, particularly when state vectors include multiple spatial fields from high-resolution climate models. In the DASH framework, these files are saved with a `.ens` extension, and the toolbox provides the `ensemble` class to facilitate memory-efficient interactions with saved state vector ensembles. We highlight the ability of the `ensemble` class to selectively load requested state vector rows, variables,

and ensemble members into memory. These features have particular utility when running (1) proxy forward models, which typically only require a small subset of ensemble data, and (2) data assimilation algorithms, as many reconstructions only target a subset of variables in an ensemble. Users can also call the `evolving` command to implement evolving offline priors (e.g. Osman et al., 2021) without loading data values to memory.

### 3.2.3 Proxy Forward Models: `PSM`, `ensembleMetadata`

After building a state vector ensemble, a common next task in paleoclimate DA is to design a forward model for each proxy record. These forward models are either used to generate proxy estimates (for offline assimilations) or provided directly as input to data assimilation algorithms (for online regimes). The `PSM` package facilitates all these tasks by providing users modular access to commonly used proxy system forward models. The actual implementation of proxy system models is beyond the scope of DASH; instead, the `PSM` package acts as a bridge to shuttle information contained with the state vector ensemble

into established proxy model codes. DASH currently supports multivariate linear models (see Hakim et al., 2016; Tardif et al., 2019; Zhu et al., 2020a), the Vaganov-Shashkin 'Lite' (VSL) tree ring model (Tolwinski-Ward et al., 2011), the `BayWATCH` suite of Bayesian foraminiferal and membrane-lipid models (Tierney and Tingley, 2014; Malevich et al., 2019; Tierney et al., 2019; Tierney and Tingley, 2018), a Palmer Drought Severity Index (PDSI) estimator (Guttman, 1991; Van der Schrier et al., 2011), and the models within the `PRYSM` Python package (Dee et al., 2015a) (Table 1). We anticipate that this list will grow

with future advances in proxy system modeling.

Users can call the `download` method to automatically download selected forward models from their respective Github repositories and add them to the MATLAB active path. The class then allows users to design `PSM` objects, which implement forward models for different proxy record with modular model parameters. Users then indicate which state vector rows hold the data needed to run each forward model; this search is facilitated by the `ensembleMetadata` class detailed in the next

paragraph. Users can then either use the `estimate` command to run the forward models over the state vector ensemble and generate proxy estimates. Users can also run the forward models over updated state vector ensembles, in order to validate proxy records against assimilation results (e.g. Tardif et al., 2019; Tierney et al., 2020b; King et al., 2021; Osman et al., 2021).



The process of running forward models on a state vector ensemble is facilitated by the `ensembleMetadata` class. This class implements objects that organize metadata along the rows and columns of a state vector ensemble. An `ensembleMetadata`

object is created whenever a user builds a state vector ensemble and can also be returned for `.ens` files and `stateVector` objects. The class can be used locate state vector rows corresponding to particular variables, spatial locations, or time sequences, and can also be used to locate specific ensemble members. A major task of `ensembleMetadata` is to locate state vector rows that correspond to proxy forward model inputs. In addition to locating specific climate variables, the class can determine which data elements are geographically closest to the location of a proxy site, which is often necessary when imple-

menting forward models. Each `ensembleMetadata` object also holds the metadata necessary to reshape state vectors back into gridded datasets. Consequently, the class is also used to reshape DA outputs back into spatial grids for post-processing and visualization.

### 3.2.4 Data Assimilation Algorithms: `kalmanFilter`, `particleFilter`, `optimalSensor`

This section describes the classes used to implement data assimilation algorithms. Each class implements objects that hold

parameters for a particular type of analysis. The object-oriented layout allows users to specify diverse algorithm parameters, while promoting the readability of analysis codes. Broadly, each class shares a similar usage syntax. Users first initialize an object for the desired algorithm and next provide required parameters. Here, required parameters typically include a state vector ensemble ($X_p$), proxy records (Y), proxy estimates ($\hat{Y}$) or forward models, and proxy error-variances or covariances (R). Users can specify any additional parameters, and then implement the algorithm using the `run` method. To support the use of large

state vector ensembles, all three DA algorithms included in DASH are optimized for both speed and efficient use of memory.

The `kalmanFilter` class contains options for offline regimes, and may also be adapted into online frameworks. The class implements an ensemble square-root Kalman filter (Andrews, 1968), which processes ensemble means and deviations separately. This separation precludes the need for perturbed observations (Whitaker and Hamill, 2002) and provides several opportunities for enhanced computational efficiency. For example, exploratory analyses can choose to only assimilate the en-

375 semble mean, which is significantly faster than updating the full ensemble. Other optimizations leverage the independence of deviation updates from the proxy records to minimize the number of computations of the Kalman Gain. The `kalmanFilter` class also supports several methods commonly used to adjust Kalman filter covariance matrices (the $\text{cov}(X, \hat{Y})$ term in Equation 5); these include covariance inflation (Anderson and Anderson, 1999), localization (Hamill et al., 2001), and blending ensemble covariances with a second covariance matrix (e.g. Valler et al., 2019). The class also permits user-specified covari-

380 ance matrices, which can be useful when climate-system covariances are poorly defined, such as for changing continental configurations in deep-time assimilations. Finally, the `kalmanFilter` class supports the use of evolving offline priors (e.g. Franke et al., 2020; Osman et al., 2021), which can be used simulate changing climate system boundary conditions while minimizing computational cost.

Naïve Kalman filter algorithms return an entire state vector ensemble in each assimilated time step, which can rapidly exceed

computer memory. Consequently, the `kalmanFilter` class includes many options for reducing the size of the outputs. Alternatives to saving full ensembles include only returning the ensemble mean, returning the ensemble mean and variance,





and returning several percentiles of the full ensemble. The class also provides support for reconstructing climate indices from assimilated spatial fields while conserving computer memory. In many cases, an assimilated spatial field is primarily used to calculate a reconstructed climate index. The full posterior of a climate index is often useful for uncertainty analysis, but spatial
fields are often too large to allow the return of full posterior ensembles. To remedy this situation, the `index` method allows users to calculate and return the full posterior of a climate index (such as global mean temperature or the Nino 3.4 index), without saving the full-field posterior ensemble. We also reiterate that users can use the `ensemble` class to only assimilate a subset of the variables in a state vector. Some variables might only be necessary to run the PSMs, and excluding these variables from the algorithm can improve both memory use and run time.

The `particleFilter` class provides an alternative algorithm to Kalman filtering. In DASH, this algorithm proceeds by weighting the state vectors (i.e. particles) in an ensemble and then computing a weighted mean across the ensemble. The primary option in the `particleFilter` class concerns the method used to determine the weights for the mean. By default, the class implements a Bayesian weighting scheme that conforms to a classical particle filter (see Van Leeuwen, 2009). However, users can instead choose to take a mean of the best $N$ particles, with the number of particles specified by the user.

The `optimalSensor` class is based on the method described by Comboul et al. (2015), which is derived from an ensemble Kalman filter framework. Rather than reconstructing climate variables over time, the algorithm instead tests the ability of a proxy record to reduce the variance of a climate metric calculated over an ensemble. Essentially, this method assesses the relative influence of individual proxy records on a reconstructed index (such as a spatial temperature mean or climate mode index). The `optimalSensor` class provides three distinct, yet related, routines to support these types of analyses. The
`evaluate` routine allows users to assess each proxy's individual ability to reduce variance in the posterior ensemble. The `run` routine implements the greedy algorithm of Comboul et al. (2015), and allows users to rank the utility of proxy sites for successive assimilation. Finally the `update` routine assesses the total variance reduced by an entire proxy network. These commands can also be combined to examine changes in proxy influence as additional records are added to a network.

Classically, the optimal sensor algorithm strictly requires proxy error-variances, which necessarily assumes that assimilated
proxy records are independent. However, the `optimalSensor` class extends the algorithm to allow for covarying proxy errors. In this case, the covarying-proxies are processed using a single block-update, effectively treating the covarying proxies as a single sensor. This is useful when assimilating gridded, spatially-covarying proxy networks and climate field reconstructions (e.g. King et al., 2022), such as drought atlases (e.g. Cook et al., 1999, 2010; Morales et al., 2020).

## 4    Examples

In this section, we provide two examples illustrating the use of the DASH toolbox. These examples are designed to demonstrate the utility of DASH for a variety of analyses over different spatial scales, time periods, and proxy networks. These examples closely mimic several existing studies in the paleoclimate DA literature (King et al., 2021; Tierney et al., 2020b; Osman et al., 2021), although we have modified the analyses at several points for brevity or to demonstrate the extended capabilities of the DASH toolbox. Numbers in parentheses refer to the line numbers in the code for each example.



## 4.1 Northern Hemisphere Summer Temperatures over the Last Millennium

Our first example illustrates a possible setup for reconstructing summer temperatures in the extratropical Northern Hemisphere over the last millennium using annually resolved proxies. This example follows the assimilations found in King et al. (2021), although for the sake of simplicity, we only assimilate a single climate model here. In this example, we integrate a network of 54 temperature-sensitive tree-ring records (Wilson et al., 2016; Anchukaitis et al., 2017) with output from the CESM1.1 Last Millennium Ensemble (LME; Otto-Bliesner et al., 2016) to reconstruct both a summer (JJA) temperature spatial field and a spatial-mean index. We generate proxy record estimates using simple linear forward models trained on the mean temperature of each site's optimal growing season. We run the assimilation using an ensemble Kalman Filter with a stationary offline prior. We also apply covariance localization for the spatial field, which we implement using a Gaspari-Cohn 2D polynomial (Gaspari and Cohn, 1999) with a 20,000 km cutoff radius. Finally, we use an optimal sensor analysis to evaluate the potential influence of each tree-ring record in the network. The results of the analysis are displayed (Figure 3) using the visualization codes in the data repository.

### 4.1.1 Organize Climate Data

The first two sections of the example (lines 6-50) illustrate using `gridfile` to organize data used in the assimilation. Here, these data consist of (1) climate model output from the CESM1.1 LME and (2) tree-ring chronologies. The climate model output contains reference height temperatures from fully-forced run #2. This output is stored across two NetCDF files and spans a 2D spatial grid over the period 850 to 2005 CE at monthly resolution. Our first step is to create a metadata object that defines the scope of this dataset (lines 12-18). Here, we choose to define spatial metadata using the latitude and longitude values stored in the NetCDF output files (lines 13-14). However, the time metadata in the NetCDF files is reported as "days since January 1, 850", which is non-intuitive for our purposes. Instead, we choose to define time metadata using MATLAB's built-in `datetime` format, which will allow us to sort time points by months and years (line 15). We also include two optional metadata attributes (the units and climate model associated with the output) to better document the dataset (line 18). We next create a `gridfile` object whose scope is defined by this metadata (line 21) and add the temperature dataset, stored in the `TREFHT` variable of the two NetCDF files, to the `gridfile` object's catalogue (lines 28-29). Finally, we apply a data transformation to the catalogue (line 32) so that loaded temperature data will be returned in units of degrees Celsius, rather than Kelvin.

In the next section (lines 35-50), we catalogue the tree-ring chronologies. These records are stored in a binary MAT-file (line 38), along with information about each proxy site. The proxy record dataset is a 2D array that spans 54 proxy sites over time at annual resolution. Here, we choose to define metadata (line 43) along the proxy-site dimension using the ID, spatial location, and optimal growing season of each site (line 42). For time metadata, we use the calendar year corresponding to each measurement (line 43). We next create a gridfile object whose scope is defined by this metadata (line 46) and add the proxy record dataset, stored in the `crn` variable of the MAT-file, to the gridfile catalogue (line 47). Finally, we indicate that -999 values in the dataset represent fill values and should be converted to NaN when loaded (line 50).





### 4.1.2 Build a State Vector Ensemble

In the next section (line 53-96), we use the `stateVector` class to design and build a state vector ensemble. We begin by
455 initializing and labeling a `stateVector` object (line 56), and then initializing variables within that state vector (lines 62-63).
Typically, a state vector will include any variables required to run the proxy system forward models, as well as reconstruction
targets. In this example, each proxy system model requires a seasonal temperature mean from the model grid point closest to
the proxy site. Thus, we first initialize variables for the temperature means of the proxy records using a different name for each
site (line 62). We also create variables for the reconstructed spatial temperature field, and the spatial-mean index (line 63), for
a total of 56 variables. All of these variables will be constructed from the monthly LME temperature output, which is indicated
by the second input in lines 62 and 63. Note that the names of state vector variables do not need to match the names of variables
stored in data source files – here, `TREHFT` – because multiple state vector variables may be derived from the same dataset.

We next specify how to select ensemble members in the state vector ensemble. In this example, we indicate that ensemble
members should be selected along the time dimension, with each ensemble member associated with a particular calendar year
(line 69). Using -1 as the first input applies this setting to every variable in the state vector. Here, we use January as a reference
point for each calendar year, but this does not imply that the variables will necessarily contain data from the month of January.
Instead, the January months are used to align variables so that the values within any given ensemble member correspond to
the same year. For example, consider two variables implementing seasonal means. One variable, MJJA, implements a seasonal
mean from May to August. The other variable, ON, implements a seasonal mean from October to November. Although the
470 two variables cover different seasonal windows, the seasonal windows for each ensemble member should be drawn from the
same year. Here, the January reference point ensures that these seasonal windows are aligned to the same year; essentially,
the variables for each ensemble member will be built using the appropriate seasonal window as indexed from the associated
January reference point. For an ensemble member that uses January 1850 as a reference point, the MJJA variable will be built
using data from May-August 1850, and the ON variable will be built using data from October-November 1850. Although the
475 two variables use different temporal spans, they collectively refer to the same year within the ensemble member. Additionally,
the state vector class will ensure that ensemble members are only selected from years that include complete temporal spans for
all variables. Continuing the example: if the temperature dataset ended in October 1900, then 1900 will never be selected as
an ensemble member because the ON variable would be missing data from November of that year. We note that users are not
confined to a given calendar year, as the months used in the seasonal window are indexed from the associated reference point.
For example, a user could implement a DJF seasonal mean by providing indices [-1 0 1], thereby creating a seasonal window
from the three monthly time steps centered on each January.

Finally, we design the variables so that each uses values from the appropriate subset of the monthly temperature dataset. For
the reconstruction targets, we use grid points from the extratropical Northern Hemisphere (line 78) and summer (June-August)
seasonal temperature means (line 79). We note that the third input in line 78 is left empty because the latitude dimension should
not be used to select different ensemble members (contrast this with the time dimension in line 69). To implement the seasonal
means, we provide the indices of months relative to each January reference point. As the reference point, each January is given





a relative index of 0; hence, a June-August mean is calculated using data values 5, 6, and 7 (monthly) time steps after each January reference point. We also specify a latitude-weighted spatial mean for the spatial-mean index (line 80). Before designing the forward-model variables, we first note that each variable uses a different seasonal average. Including the full spatial field for multiple different seasonal windows would result in an unnecessarily large state vector, so we first use the `closest.latlon` utility to locate the model grid point closest to each proxy site (line 86). We then design each forward-model variable to consist of the site-specific seasonal temperature mean at that single grid point (lines 87-93). At this point, we have finished designing the state vector, and proceed to build an ensemble with 1000 members (line 96). In this example, we save the built ensemble to a `.ens` file. Although the `stateVector` class can also return ensemble directly as output, we generally recommend saving to file, because this allows the DASH toolbox to use computer memory more efficiently.

### 4.1.3 Proxy Forward Models

The next section (lines 99-118) uses the `PSM` package and `ensembleMetadata` class to design proxy forward models and run the models on values stored in the state vector ensemble. The outputs of these forward models are the proxy estimates used to compare state vector ensemble members to observed proxy records in assimilation algorithms. We begin by using the `PSM` package to create simple, linear forward models for each proxy site (line 109). The coefficients for each model are calibrated to mean temperature over the optimal growing season at each proxy site. Determining forward model coefficients is beyond the scope of this example, but King et al. (2021) compute these values by regressing the proxy records against an instrumental temperature dataset. After designing each model, we next indicate the state vector row that corresponds to the inputs for each model (lines 113-114). Finally, we use the `estimate` command to run the forward models on the ensemble and generate the proxy estimates (line 118).

### 4.1.4 Kalman Filter

In this section (lines 121-174), we use the `kalmanFilter` class to implement an ensemble Kalman filter and reconstruct summer temperatures. We first initialize and label a `kalmanFilter` object, which will store the parameters used to run the assimilation (line 124). The mandatory parameters for an ensemble Kalman filter are (1) a prior ensemble, (2) proxy records, (3) proxy estimates, and (4) proxy error covariances or variances, and we provide these parameters to the `kalmanFilter` object in lines 130, 133, 135, and 139. Determining proxy error variances is beyond the scope of this example, but King et al. (2021) compute these values by running the proxy forward models on an instrumental temperature dataset and comparing the resulting proxy estimates to the real proxy records. In this example, we also implement covariance localization. To accomplish this, we first calculate localization weights for the ensemble and proxy sites (line 144) and then provide these weights as parameters to the kalmanFilter object (line 145).

To illustrate the flexibility of the DASH architecture, we also demonstrate a second method for reconstructing the spatial-mean summer temperature index (line 153). This method allows the user to calculate an index from the posterior of a spatial field, without saving the (often very large) spatial field posterior. To further conserve memory, we also indicate that the filter should only record the variance and percentiles of the posterior ensemble (lines 156-157), rather than the much larger full





posterior. Finally, we run the Kalman filter algorithm for the analysis and return the mean, variance, and posterior mean and percentiles of the target reconstruction variables (line 160). We note that the reconstructed spatial field is organized as a state vector, but many mapping functions operate on spatial matrices, rather than vectors. Hence, to facilitate display of the reconstructed spatial field, we regrid the posterior to the spatial dimensions of the original climate model output (lines 164-165). We also extract the assimilated spatial temperature mean, which is the final element along the state vector (line 168), and the alternative spatial mean, which was calculated from the updated spatial field (line 170).

Figure 3 illustrates the results of this assimilation. The upper panel compares the reconstructed indices obtained using the two different methodologies: the blue line depicts the index obtained by assimilating the temperature spatial mean directly in the state vector, and the red line depicts the index calculated from the updated (posterior) spatial field. The lower left and lower center panels display an example reconstructed spatial field from 1850 CE along with an uncertainty quantification based on the variance of the field's posterior ensemble. Notably, the spatial indices calculated using the two different methods are not identical. In brief, this discrepancy occurs because (1) the index calculated from the posterior field (in red) is sensitive to spatial heterogeneity in the Kalman filter updates, and (2) the directly assimilated index (in blue) is less sensitive to the proxy records than are individual spatial sites. The causes and implications of this behavior are discussed in greater detail in Section 5.3.

### 4.1.5 Optimal Sensor

In the final section (lines 177-194), we use an optimal sensor framework to evaluate the influence of each proxy on the reconstructed spatial-mean index. Analogous to the `kalmanFilter` object of the previous section, here we will use an `optimalSensor` object to organize parameters for the analysis. The required parameters for an optimal sensor are (1) a sensor metric, (2) proxy estimates, and (3) proxy error variances or covariances. After initializing and labeling the `optimalSensor` object (line 180), we set the extratropical summer temperature index as the sensor metric (lines 183-184) and also provide proxy estimates and error-variances (lines 187-188). With these parameters set, we then use the optimal sensor to evaluate the power of each proxy for reconstructing the spatial-mean index (lines 191).

Figure 3 (lower right) displays the results of this analysis. Here, the ability of a proxy to reduce variance responds to two factors: the covariance of its estimates with the modeled spatial-mean index, and its uncertainty values (R), which represent the accuracy of its forward model. Thus, the proxies with the greatest ability to reduce variance are characterized by more accurate forward models and stronger covariance with the spatial-mean index.

### 4.2 Global Sea Level Pressures at the Last Glacial Maximum

Our second example illustrates a setup for reconstructing global sea level pressures from the Last Glacial Maximum (LGM) to present. This example is inspired by Osman et al. (2021) with several modifications. First, we assimilate global sea level pressures rather than sea surface temperatures (SSTs) in order to demonstrate the reconstruction of climate variables not directly sensed by the proxy network. For the sake of simplicity, we also limit the proxy network to the alkenone $U_{37}^{K'}$ and $\delta^{18}O$ of planktic foraminifera SST proxies, neglect spatial variations in proxy seasonal sensitivities, and reconstruct spatial fields on a 3,000-year time step. In this example, we integrate a network of $U_{37}^{K'}$ and $\delta^{18}O$ sediment records with output from the isotope-





enabled Community Earth System Model (iCESM1.2; Brady et al., 2019; Tierney et al., 2020b; Zhu et al., 2017; Stevenson et al., 2019). We generate proxy record estimates using the `BayFOX` (Malevich et al., 2019) and `BaySPLINE` (Tierney and

555 Tingley, 2018) forward models. We conduct the assimilation using an ensemble Kalman filter with an evolving offline prior and also implement a proxy-validation analysis. The results of this analysis are displayed in Figure 4 using the visualization codes in the data repository.

### 4.2.1 Organize Climate Data

Similar to Example 1, the first two sections again use the `gridfile` package to organize climate data. Here, the data consists

of (1) climate model output from iCESM binned to 50-year monthly climatologies, and (2) $U_{37}^{K'}$ and $\delta^{18}O$ proxy records. The climate model output includes variables for the sea-level pressure (SLP) reconstruction target, as well as sea surface temperatures (SSTs) and $\delta^{18}O_{sw}$, which are used to run the proxy forward models. The climate variables reflect mean monthly values averaged over 50-year intervals in order to more closely match the multi-decadal averages captured by the proxy records (Tierney et al., 2020b; Osman et al., 2021). This output includes sixteen 50-year averages for each of the nine 3,000-year

intervals from the LGM to present, for a total of 144 possible ensemble members. The data for the variables are stored in three separate NetCDF files. The SLP variable is provided on a rectilinear atmosphere grid, and the creation of its gridfile catalogue (lines 13-18, 27, 32) follows the process outlined in section 4.1.1. By contrast, the SST and $\delta^{18}O_{sw}$ variables are sourced from the ocean component of the model, which uses a tripolar coordinate system. Tripolar datasets typically include dimensions for both latitude and longitude, but spatial metadata is not fixed for any given element of either dimension. For example, the

latitude value at $(\text{latitude}_j, \text{longitude}_k)$ is not the same as the latitude value for $(\text{latitude}_j, \text{longitude}_{k+1})$. Consequently, the dataset describes values at distinct (latitude, longitude) points, rather than values on a rectilinear (latitude x longitude) grid. The `gridfile` class requires fixed metadata values along each data dimension, so we define the metadata for SST and $\delta^{18}O_{sw}$ using unique spatial sites (lines 20-24), rather than a rectilinear latitude x longitude format. Note on lines 33 and 34 that two dataset dimensions are associated with the site spatial dimension. This syntax merges the latitude and longitude dimensions

in the `gridfile` catalogue and treats them as a single spatial dimension. We next use `gridfile` to catalogue the proxy records (lines 37-49). Here, the proxy records are stored in a binary MAT-file, along with metadata describing the records. This metadata includes the ID, spatial coordinates, proxy type ($U_{37}^{K'}$ or $\delta^{18}O$), and foraminiferal species associated with each record (line 43).

### 4.2.2 State Vector Ensemble and Evolving Prior

We next design and build the state vector ensemble for the LGM assimilation (lines 52-80). We begin by initializing a `stateVector` object with three variables (lines 55 and 60). The first variable, SLP, is the reconstruction target; the other two variables, SST and $\delta^{18}O_{sw}$, are required to run the proxy forward models. We next indicate that ensemble members should be selected from different points in time with each ensemble member associated with a particular 50-year average and we specify January as the reference point (line 65). In this example, we target annual SLP values. Since we are neglecting spatial



variations in proxy seasonal sensitivities, we also require annual SST and $\delta^{18}O_{sw}$ values as proxy forward model inputs. Thus, we use an annual mean for each of the three variables (line 69).

We note that, unlike Example 1, we do not design variables for the individual proxy records; instead, we include the entire spatial field for each climate variable used by the forward models. This syntax simplifies the code but results in a larger state vector. We elect to use this syntax here in order to improve code clarity and also demonstrate the flexibility of the DASH

architecture. However, other applications should compare the benefits of code clarity with greater memory use when choosing a syntax. Finally, we build a state vector ensemble using all available ensemble members (line 73). We select ensemble members sequentially in order to facilitate the creation of an evolving prior. This orders the ensemble members so that the sixteen 50-year averages for each 3,000-year interval are all in succession. We next use the `evolving` command to implement an evolving prior for the different 3,000-year intervals (line 80). For this command, the columns of the `members` variable indicate which

ensemble members should be used for each evolving prior. Here, each prior is built using the sixteen 50-year averages for one of the nine 3,000-year intervals.

### 4.2.3 Proxy Forward Models

We next build and run proxy forward models on the state vector ensemble in order to generate a set of proxy estimates. Here, we use the `BaySPLINE` and `BayFOX` Bayesian forward models for $U_{37}^{K'}$ and $\delta^{18}O$, respectively. We begin by using

the `download` command to download the models from their respective Github repositories and add them to the MATLAB active path (lines 86-87). We next design a forward model for each proxy record using the model appropriate for each proxy's type (lines 90-120). For the `BaySPLINE` model, we locate state vector rows corresponding to the SST values from the climate model grid point closest to each proxy record (lines 101-104). The `BayFOX` model is calibrated to different foraminiferal species, so we initialize each model with the species of the associated proxy record (line 109). We then locate both SST and

$\delta 18O_{sw}$ values, again at the closest climate model grid point (lines 112-114). For the purposes of documentation, we also label each forward model with the ID of the associated proxy record (line 118). Finally, we run the forward models on the evolving state vector ensemble using the `estimate` command (line 124). In addition to proxy estimates, the `BaySPLINE` and `BayFOX` models calculate proxy error-variances, provided as the second output, which we use to define R.

### 4.2.4 Kalman Filter and Proxy Validation

We next implement the Kalman Filter analysis (lines 127-149). We first initialize and label a `kalmanFilter` object (line 130) and then provide the required algorithm parameters (lines 134-137). To conserve memory, we only return the mean and variance of the posterior ensemble (line 141). As in Example 1, we regrid the reconstructed spatial field to the dimensions of the original climate model to support visualization and post-processing (lines 145-146; Figure 4). Unlike Example 1, we include all of the climate variables needed for the proxy forward models in the prior. This allows us to run the proxy forward models

on the reconstruction and generate proxy posterior estimates. We can then compare these estimates to the real proxy records as a basic assessment of reconstruction skill (Figure 4). We implement this process by applying the `estimate` command to the posterior (line 155). For the sake of brevity, we only implement a simplified proxy validation in this example. In practice, DA





applications should validate the reconstruction using proxies withheld from the assimilation (e.g. Tierney et al., 2020b; Osman et al., 2021; King et al., 2021) so that assimilated proxies do not inform the skill of their own validation values.

### 4.3 Additional Considerations

The examples presented above touch upon many aspects of paleoclimate DA workflows but cannot be exhaustive. For the sake of brevity and clarity, we have neglected several considerations common in DA applications. One particular step we have omitted is the determination of proxy uncertainties (R in equations 3, 5, and 6). In some cases, proxy uncertainties (R) may be provided by the proxy forward models (as in Example 2) or from the calibration of the forward models (e.g. Tardif et al., 2019; King et al., 2021). Another potential approach involves running the forward models on instrumental data and comparing the resulting proxy estimates to the real proxy records (e.g. King et al., 2021, 2022). However, we note that these approaches are not applicable to all analyses, so users may need to develop additional methods to estimate proxy uncertainties. For example, methods that estimate proxy error-variances (e.g. Tardif et al., 2019; Tierney et al., 2020b; King et al., 2021) implicitly assume the independence of proxy uncertainties. However, this assumption may not hold when proxy records are strongly correlated or sensitive to the same local factors. When this occurs, proxy error-covariances should be used in place of error-variances (see King et al., 2022, for an example). We also discuss additional issues common to many paleoclimate applications in the section below.

## 5 Warnings and Best Practices

While it is not possible to detail all the issues that can occur when using DA for paleoclimate reconstructions, here we mention several cautions and suggestions for best practices. Along with methodological considerations, DA users should be aware of the limitations of both the proxy data and prior modeled climate states. In other words, simply running an assimilation code does not guarantee that a reconstruction is scientifically valid, and potential DA users should understand the tradeoffs and limitations of DA methods when designing a reconstruction. In this section, we present several major challenges that may be encountered in paleoclimate DA and outline approaches to mitigate or recognize their effects. This list is by no means exhaustive, and we strongly recommend that potential DASH users first familiarize themselves with the paleoclimate DA literature and also evaluate their reconstructions for sensitivity to the assumptions and input data.

### 5.1 Temporal Variability

A major issue when using an ensemble Kalman filter with a static prior (e.g. Steiger et al., 2014; Hakim et al., 2016; Dee et al., 2016; Tardif et al., 2019; Steiger et al., 2018; Neukom et al., 2019; Zhu et al., 2021a; King et al., 2021, 2022) is that the proxy network's size and composition – and changes to these properties over time – can directly alter the temporal variability of the reconstruction. Essentially, we observe that variability is artificially reduced as the proxy network becomes smaller. It is common for the sample size of a proxy network to change over a reconstructed time period. When this occurs, then the



variability of the reconstruction will be non-stationary and relative climate variability may not remain consistent over the span of the reconstruction.

This effect occurs because a static prior implies zero temporal variability as an *a priori* assumption in the absence of proxy information. Consider a "no-information" case, in which a static prior is assimilated with an empty proxy network. Since the proxy network is empty, the prior ensemble will not receive any updates, and the reconstruction will be the mean of the prior in every time step. Since the prior is identical in every time step, the reconstruction will consist of a constant value over time and will exhibit no temporal variability. With the addition of a proxy record to the network, the prior will begin to receive updates,

and the reconstruction will begin to gain temporal variability. Each subsequent record added to the proxy network increases the ability of the method to move the reconstruction away from the prior mean, and so reconstruction variability will increase with the size of the proxy network. This behavior is by design: in the absence of additional information, the prior provides the best estimate of the mean state of the climate system. However, it creates complications for paleoclimate interpretations. We note that this effect is most severe for smaller proxy networks and at spatial points informed by a limited number of proxy records.

Because of this effect, it is essential that assimilations using static priors account for the effects of proxy network composition on temporal variability. Variance adjustment methods are common in other approaches to paleoclimate reconstruction (e.g. Cook et al., 1999; Esper et al., 2005; Frank et al., 2007; Anchukaitis et al., 2017), and King et al. (2022) provide an example for how this can be accomplished for DA applications. Alternatively, evolving priors can mitigate the variance issue (e.g. Tierney et al., 2020b; Osman et al., 2021) by removing the *a priori* assumption of zero temporal variability. However, we

caution that evolving priors could still exhibit a variance dampening effect when the variability between reconstruction time steps and the state of the evolving priors is dominated by internal climate variability.

## 5.2    Climate model biases

A second major concern for paleoclimate DA concerns the effects of climate model biases on assimilated reconstructions. In this discussion, we find it useful to distinguish between (1) biases in the mean state, and (2) climate model covariance biases.

Mean state bias refers to the systematic tendency of a simulated variable to be too high or too low compared to observations. Covariance bias refers to errors in the linear relationship between climate variables at different spatial points, or between different variables. Essentially, these are biases in the teleconnection patterns associated with various climate phenomena. Since the model prior covariance determines how information propagates from a proxy network to distal parts of a climate field, differences between the real and modeled climate system covariance will cause errors in the assimilation. No climate

model can match the complexity of the real Earth system and so all climate models necessarily include some degree of error.

An additional consequence of climate model biases concerns method testing and proof-of-concept studies for paleoclimate DA. Typically, these studies rely on pseudo-proxy frameworks (Smerdon, 2012), in which climate model output is used to simulate a set of proxy records. These pseudo-proxy records are designed to mimic a real proxy network and can be used to reconstruct the climate model output. Unlike the real past climate history, the climate model output is fully known and so these

experiments provide an opportunity to assess assimilation skill. Due to the complexity of skill assessments, it can be tempting to use the same climate model to both generate the pseudo-proxies and build the assimilation prior. However, we caution that



this framework represents an unrealistic "perfect-model" design, in which the climate model used for assimilation perfectly describes the target climate system. Although perfect-model experiments have their uses, climate model biases represent a major source of error in paleoclimate DA (Dee et al., 2016; King et al., 2021) and DA users should account for these biases to accurately quantify DA skill. Ultimately, "biased-model" experiments, which use different climate models to generate pseudo-proxies and build the assimilation prior, are necessary for accurate method testing. We also note that the exact nature of climate model biases will vary by the choice of model and the specific target climate variable(s), and so an ensemble of different biased-model tests is often necessary to capture the full effects of climate model biases.

Deleterious effects in real assimilations also occur when the inputs to the proxy forward models exhibit mean state biases. For example, consider using the VS-Lite tree-ring model (Tolwinski-Ward et al., 2011) to assimilate a climate model with a persistent cold bias. VS-Lite includes a temperature threshold based on absolute Celsius units; at temperatures below this threshold, VS-Lite assume no growth occurs and produces a proxy estimate of zero. As a result, a climate model with a cold bias may consistently fall below this threshold, causing VS-Lite to estimate a null proxy record. In this case, as a consequence of the mean state bias in the climate model, VS-Lite would assume that trees cannot grow at a location where they do grow in reality, and this error would degrade the reconstruction. More generally, mean state biases propagate through the forward models to the proxy estimates and thereby influence the comparison of the ensemble members to the real proxy records. In some cases, these biases can cause artificial trends in a reconstruction. Essentially, the assimilation draws reconstructed variables unilaterally in the direction of less biased mean values. Although this does indeed improve the final estimate of a variable's value, this behavior is mixed with the variable's reconstructed temporal evolution and causes an artificial trend.

Some mean state biases can be addressed by the process of bias correction used in other disciplines and applications (e.g. Wang and Robertson (2011); Zhao et al. (2017); Cannon et al. (2015); Cannon (2018); Galmarini et al. (2019), and see Steiger et al. (2018) for a DA example). When appropriate, users can alternatively avoid the effects of mean state biases by providing climate anomalies to the proxy forward models, rather than absolute values (e.g. Tardif et al., 2019; King et al., 2021, 2022). This is often appropriate for assimilations that rely on linear proxy forward models or forward models not dependent on absolute units. If using priors from multiple climate models, users may also need to avoid or account for time periods when climate models strongly differ, as strongly differing climate representations can act analogously to mean state biases. For example, the instrumental era is often not suitable for computing climate model anomalies for long preindustrial and last millennium simulations, because the climate response to recent anthropogenic influences can vary across models during the relatively short historical period. By contrast, anomalies assessed relative to the entire pre-industrial period are typically more stable.

Covariance biases are perhaps the more challenging issue to deal with since they bias the propagation of information from the proxy records to the reconstruction targets and do not present simple fixes. Multivariate bias correction methods may provide a solution to this issue (e.g. Cannon, 2018; Vrac, 2018; Galmarini et al., 2019), but these methods have thus far seen little use in paleoclimate DA contexts. Instead, a more common solution is to assimilate a multi-model ensemble (Parsons et al., 2021; King et al., 2021, 2022). Users may enact this using a single multi-model prior (e.g. Parsons et al., 2021; King et al., 2022), or by performing an ensemble of assimilations using different single-model priors (e.g. King et al., 2021). When possible, we recommend the use of multi-model priors. These priors are supported in the DASH framework, and they limit the



effects of covariance biases by down-weighting covariance patterns that disagree across different models. We also note that this down-weighting may in part contribute to spatial heterogeneity in Kalman filter updates, which we discuss in detail in the next section.

### 5.3 Physically inconsistent reconstructions

Both the particle filter and Kalman filter frameworks assume that all state vector variables and proxy estimates follow a Gaussian distribution; however, not all climate variables meet this criterion. Thus, DA users should take care to transform non-Gaussian variables into an approximately Gaussian space before assimilation. Failing to take this step can result in unrealistic or nonphysical reconstructed values. This is often relevant when assimilating variables distributed near the lower bounds of

725 their domains. For example, precipitation variables typically have a high probability near zero, yet cannot fall below zero, and this results in a strongly non-Gaussian distribution. Because of this, raw precipitation values are not suitable for assimilation and using them can cause the method to return non-physical negative precipitation values. Users should therefore transform precipitation into an approximately Gaussian shape before assimilation. The reverse transformation can then be applied to the assimilated variables in order to obtain reconstructed precipitation. Possible transforms for variables near a lower bound

include the extended Box-Cox and log transforms (Wang et al., 2012), and the logit transform may be appropriate for variables on a finite interval (such as any variable that represents a percentage). Ultimately though, the most appropriate transforms will vary by application (Wang et al., 2012).

We also emphasize that the DA algorithms described in this paper do not conserve physical properties like mass or energy. Consequently, assimilated reconstructions are not bound by the governing equations inherent to the climate models used to

735 generate a prior ensemble and can produce physically inconsistent values. In some cases, this may mean that assimilated fields are not suitable for providing boundary conditions for climate model simulations. Unrealistic values can also arise when individual proxy records are given excessive weight in the Kalman filter. When the magnitudes of proxy weights are too large, small proxy innovations can result in drastically large updates to assimilated climate variables. This issue most commonly occurs when proxy uncertainties ($\mathbf{R}$) are severely underestimated. For example, in Example 1 our proxy uncertainties incorporate

both forward-model errors and non-climatic noise in the proxy records. However, if we neglect these effects and compute $\mathbf{R}$ using only the uncertainties inherent in measuring tree-ring variables (which are vanishingly small), the resulting Kalman filter updates alter the assimilated temperature field by thousands of degrees Kelvin, a clearly unrealistic result. This behavior underscores the importance of correctly incorporating multiple sources of error when quantifying proxy uncertainties. Although DA methods that conserve physical properties do exist, these methods have seen little use in paleoclimate contexts, likely due

to the prevalence of offline configurations.

A related issue concerns the spatial heterogeneity of Kalman filter updates, which can also result in physically inconsistent behavior. When assimilating spatial climate fields, the magnitudes of Kalman filter updates often vary unevenly across different spatial points. The magnitude of the update at a given spatial point is proportional to that point's covariance with the proxy estimates, so distant spatial points that covary less strongly with the proxy network will receive smaller updates. As a result,

reconstructed values at distant sites tend to remain closer to the prior ensemble mean and exhibit lower temporal variability





than sites closer to the proxy network. This lower variability is not a real climate phenomenon, but rather a consequence of the Kalman filter method, which is designed to estimate mean states rather than temporal variability. However, we also note that the variance of the posterior ensemble is available for users to assess the uncertainty resulting from smaller updates.

This spatial heterogeneity also has consequences for reconstructing large-scale climate indices, such as those used to char-
755 acterize first-order climate modes and spatial averages. These indices are typically computed using values from multiple points in a spatial climate field; however, the uneven application of Kalman Filter updates to different spatial points can skew the calculation of these indices. For example, consider the Southern Annular Mode (SAM): one index commonly used to measure the SAM's phase is defined using the gradient of zonal mean sea level pressures between 40°S and 65°S (Gong and Wang, 1999). Consider an assimilation that uses a proxy network primarily located near 65°S. Because of the location of the proxy
network, spatial points near 65°S will receive larger updates than those near 40°S; by contrast, points near 40°S will be less altered and will remain close to the mean of the prior. As a consequence of this effect, a SAM index determined from the posterior spatial field using this network might only reflect changes to values at 65°S, thereby failing to assess changes at the northern end of the gradient. Thus, when reconstructing climate indices from posterior spatial fields, it is essential for DA users to demonstrate the homogeneity of update magnitudes at the spatial points used to calculate the index. An alternative approach
to reconstructing climate indices is to include the climate index directly in the state vector, which precludes the issue of spatial heterogeneity. A tradeoff of this approach is that proxy records will covary less strongly with large-scale indices than with local climate variables, and so reconstruction uncertainty may remain higher overall. However, in the case of spatial heterogeneity, we emphasize that higher uncertainties are preferable to a physically implausible reconstruction.

## 6 Past Applications and Future Development

Because of its flexibility, earlier versions of the DASH toolbox have already been used to implement several paleoclimate reconstructions, ranging across a variety of time scales and reconstruction targets. Tierney et al. (2020b) used a DASH prototype to reconstruct global temperatures at the Last Glacial Maximum using a large proxy network of geochemical SST proxies and model output from iCESM1.2. King et al. (2021) used the toolbox to reconstruct summer temperatures in the extratropical Northern Hemisphere over the last millennium by integrating a temperature-sensitive tree ring network with an ensemble of
climate model simulations. Osman et al. (2021) used DASH to produce a full-field reconstruction of surface temperatures from the Last Glacial Maximum to present. Rather than conducting a field reconstruction, King et al. (2022) targeted a climate mode index and reconstructed the Southern Annular Mode over the Common Era using a southern hemisphere proxy network, drought atlases, and a multi-model ensemble. In a deep-time application, Tierney et al. (2022) used DASH to produce a temperature field reconstruction of the Paleocene-Eocene Thermal Maximum. In all of these studies, DASH was used to
implement the assimilation workflow.

DASH is an active project and we anticipate continued developments to the toolbox. Currently, we have three major areas of focus for future improvement. First, we note that proxy system modeling is an area of active research. We anticipate the devel-
opment of new proxy models and recognize the need to incorporate these future models into the DASH framework. Thus, we



are continuing to expand the PSM package to include a more diverse array of published forward models. Furthermore, DASH includes templates for adding proxy forward models, thereby allowing users to incorporate new models into the toolbox as the need arises. Second, we intend to expand DASH's support of online assimilation algorithms. DASH has primarily been used to implement offline assimilation regimes, and this has influenced the development of the toolbox. We note that DASH already provides a scaffold for online assimilations, as the routines in the toolbox can be used to update climate model output before reinitializing a climate model externally. However, future development will include adding explicit wrappers to commonly used Earth system emulators and models of varying complexity. For example, SPEEDY-IER (Dee et al., 2015b) and linear inverse models (Perkins and Hakim, 2020) have been used to implement assimilations, and both are targets for further development of DASH. Third, we recognize that DASH's reliance on MATLAB precludes a fully open-source toolbox. Although the source code for DASH is public, the toolbox will not be accessible to users lacking a MATLAB license. Consequently, long term we aim to port the toolbox to a native Python and/or Julia package.

## 7  Conclusions

In this paper, we describe the features and foundations of DASH, a MATLAB toolbox supporting paleoclimate data assimilation. The toolbox is designed for scripting and command-line use and helps implement common tasks in paleoclimate data assimilation workflows. Broadly, these include integrating data stored in different formats, designing state vector ensembles, running proxy system forward models, and implementing computationally-efficient data assimilation algorithms. The toolbox provides an interface for external, proxy-system models commonly used in the paleoclimate literature. Data assimilation algorithms in the toolbox include ensemble Kalman filters (both offline and online regimes), particle filters, and optimal sensor analyses. The package is highly flexible and is designed for general paleoclimate data assimilation, rather than any particular DA analysis. As a result of this flexibility, DASH has already been used to implement published paleoclimate reconstructions for a variety of time scales, spatial regions, and proxy networks.

*Code and data availability.* Releases of the DASH toolbox are available on DASH's Github repository (https://github.com/JonKing93/DASH/releases), on MATLAB FileExchange (https://www.mathworks.com/matlabcentral/fileexchange/120453-dash), and in the MATLAB Add-On Explorer. The DASH source code is also available on the Github repository (https://github.com/JonKing93/DASH). The input data sets, DASH 4.2.0 release, and visualization codes used in the examples of this paper are available in a public Zenodo repository (https://zenodo.org/record/7545722).

*Author contributions.* J.K. wrote the source code for the DASH toolbox. Program conceptualization, the feature set, and the overarching research goals were developed by J.K., K.J.A. and J.E.T. All authors tested the toolbox, suggested additional features and improvements, and provided feedback on implementation, documentation, and tutorials. J.K. and M.O. designed the usage examples. All authors wrote the paper.



*Competing interests.* The authors declare no competing interests.

*Acknowledgements.* We thank Jessica Badgeley, Steven Malevich, and Feng Zhu for testing early versions of DASH, suggesting features, and for fruitful discussions about implementing paleoclimate data assimilation. We also thank Dave Meko, Sylvia Dee, and Suz Tolwinski-Ward for developing and publishing proxy forward model codes used by DASH. We thank the NCAR modeling group for producing and publishing climate model output used in our examples. The development of DASH was supported by grants from the US National Science Foundation (AGS-1803946, to K.J.A.), grant 2016-015 from the Heising-Simons Foundation (to J.E.T.), and the David and Lucile Packard Foundation (to J.E.T.).



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

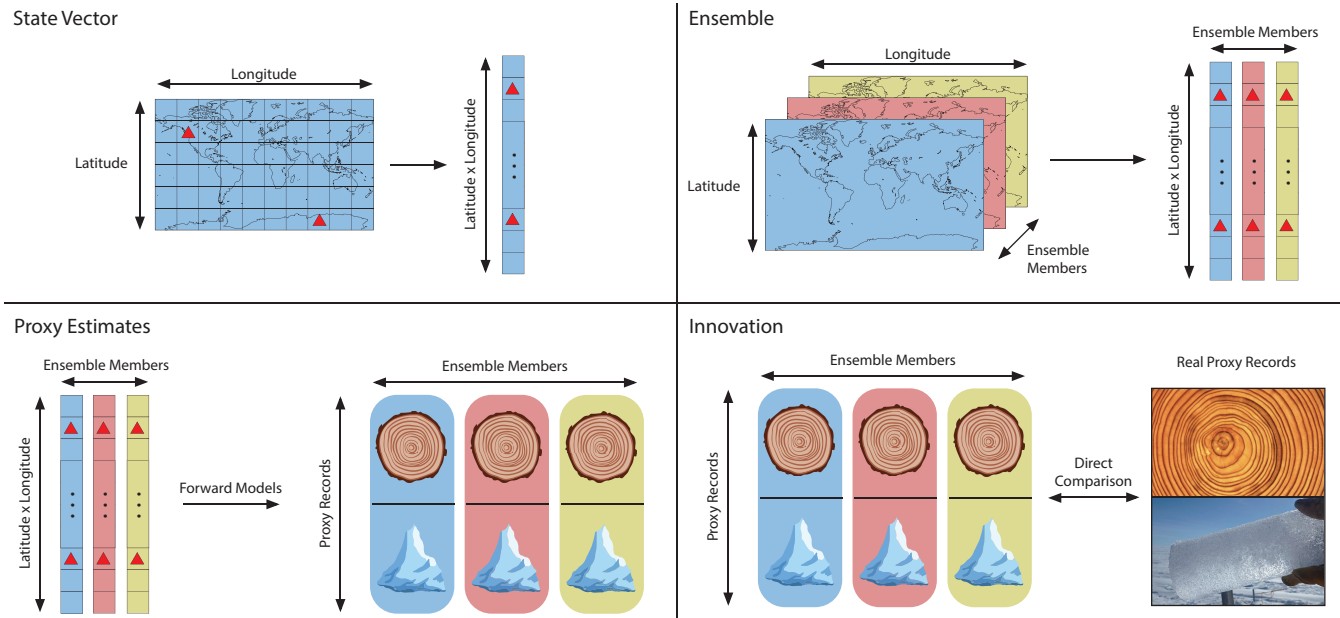

**Figure 1.** Illustration of common tasks and vocabulary for paleoclimate data assimilation. Top left: Gridded climate model output is reshaped into a *state vector*. Red triangles indicate the location of proxy records. Top right: Multiple climate model outputs are reshaped into state vectors and concatenated into an *ensemble*. Bottom left: Forward models are applied to each state vector and used to generate *proxy estimates* for each proxy record. Bottom right: Proxy estimates are compared directly to the real proxy records. The difference between the estimates and the real records is the *innovation*.





**Figure 2.** Flowchart illustrating DASH components and their uses within the context of paleoclimate data assimilation workflows.





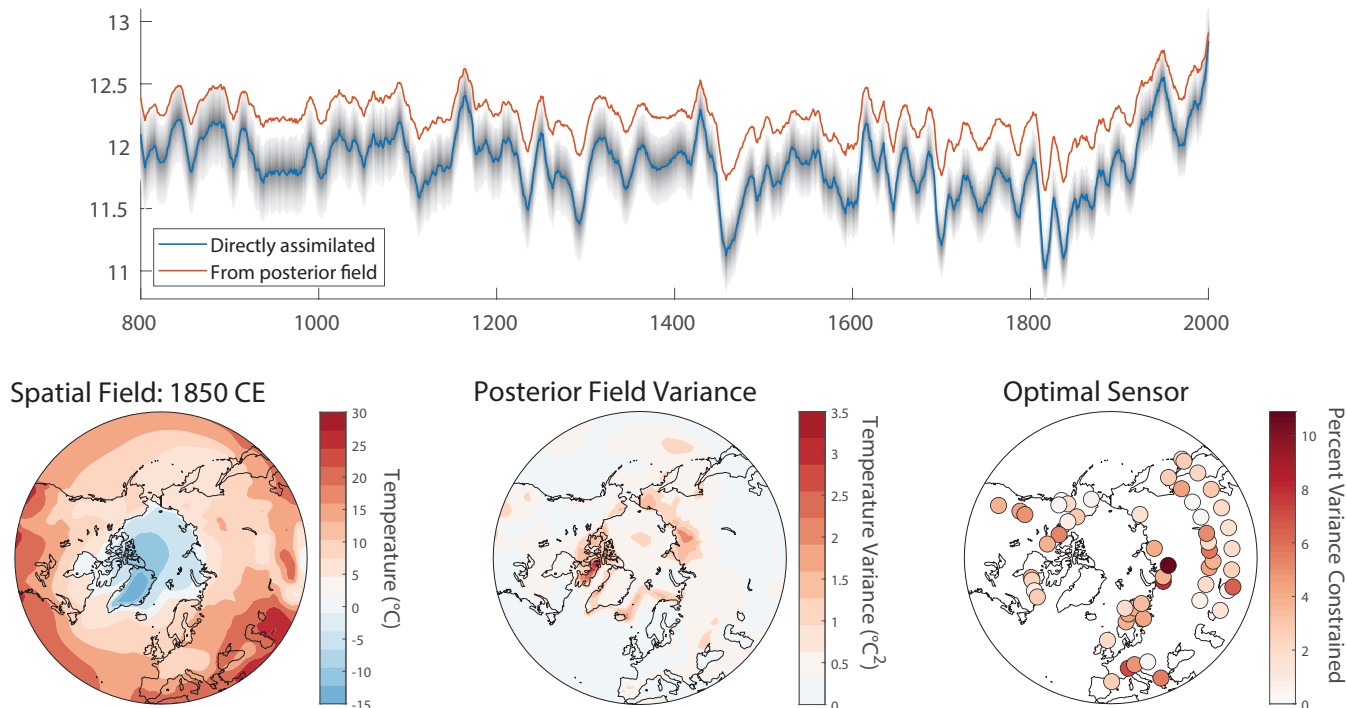

**Figure 3.** Results from Example 1, the NTREND assimilation. Top: Reconstructed mean extratropical summer (June-August) temperatures. The blue line shows the reconstructed index when the index is assimilated directly in the state vector. The red line shows the index calculated from the posterior spatial field. Grey shading indicates the 5-95% confidence level for Index 1. Lower left: The reconstructed summer-temperature spatial field in the year 1850 CE. Lower center: The variance of the posterior spatial field in the year 1850 CE. High variance indicates greater uncertainty in the reconstructed spatial field. Lower right: Results of the optimal sensor analysis. Circles indicate the locations of the NTREND tree-ring records. The color of each circle indicates the percent variance of the reconstructed index that is constrained by assimilating each NTREND site individually.



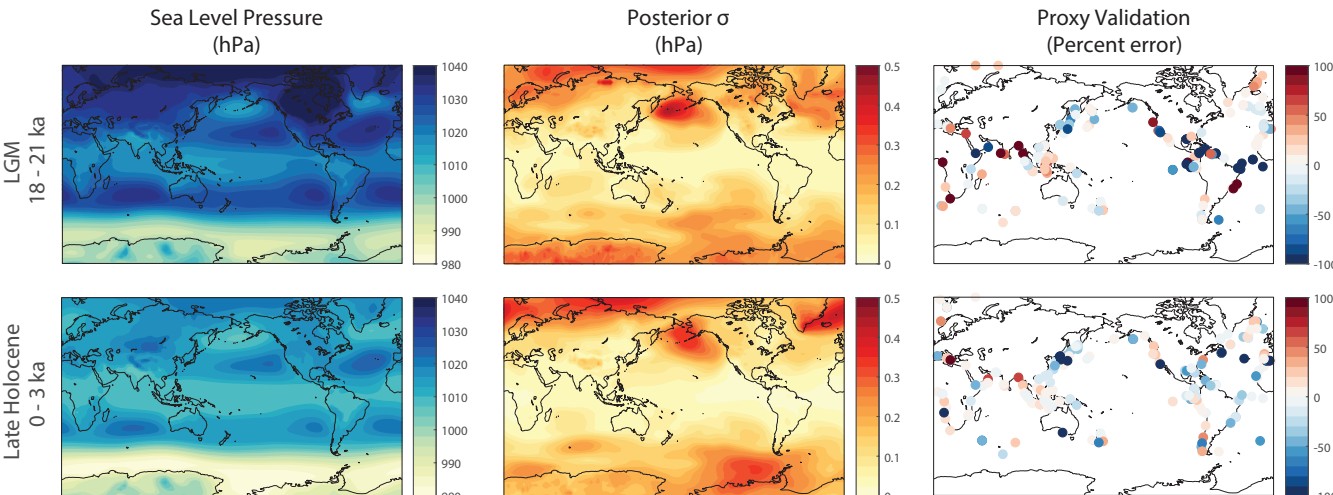

**Figure 4.** Results from Example 2, the LGM assimilation. Upper row shows results for the Last Glacial Maximum (18-21 ka); lower row shows results for the late Holocene (0-3 ka). From left to right, columns display reconstructed sea level pressure fields (hPa), the standard deviation across the posterior ensembles for each reconstructed field (hPa), and the absolute percent errors from the proxy validation.



**Table 1.** Proxy forward-models currently supported by DASH.

| Model | Description | Citation or Authors | Github Repository |
|---|---|---|---|
| BayFOX | Bayesian model of planktic foraminiferal $\delta^{18}O_c$ | Malevich et al. (2019) | jesstierney/bayfoxm |
| BayMAG | Bayesian model of planktic foraminiferal Mg/Ca | Tierney et al. (2019) | jesstierney/BAYMAG |
| BaySPAR | Bayesian model for TEX'86 | Tierney and Tingley (2014) | jesstierney/BAYSPAR |
| BaySPLINE | Bayesian model for UK'37 | Tierney and Tingley (2018) | jesstierney/BAYSPLINE |
| Identity | The identity function | | DASH built-in |
| Multi-variate Linear | General multi-variate linear forward models | | DASH built-in |
| PDSI | Palmer Drought-Severity Index estimator | Dave Meko, Jonathan King | JonKing93/pdsi |
| PRYSM Cellulose | Cellulose $\delta^{18}O$ | Dee et al. (2015a) | sylvia-dee/PRYSM |
| PRYSM Coral | Coral $\delta^{18}O$ | Dee et al. (2015a) | sylvia-dee/PRYSM |
| PRYSM Ice-Core | Ice-core $\delta^{18}O$ | Dee et al. (2015a) | sylvia-dee/PRYSM |
| VS-Lite | Vaganov-Shashkin Lite model of tree-ring width | Tolwinski-Ward et al. (2011) | suztolwinskiward/vslite |



### Example 1: Northern Hemisphere Summer Temperatures over the Last Millennium

```matlab
1   %% Example 1: NTREND Assimilation
2
3   % Reset the random number generator so all examples are reproducible
4   rng('default');
5
6   %% gridfile: Organize climate model output
7
8   % List of climate model output files
outputFile1 = 'b.e11.BLMTRC5CN.f19_g16.002.cam.h0.TREFHT.085001-184912.nc';
outputFile2 = 'b.e11.BLMTRC5CN.f19_g16.002.cam.h0.TREFHT.185001-200512.nc';
12  % Define metadata that spans the climate model output dataset
lat = ncread(outputFile1, 'lat');
lon = ncread(outputFile1, 'lon');
time = datetime(850,1,15):calmonths(1):datetime(2005,12,15);
metadata = gridMetadata('lat', lat, 'lon', lon, 'time', time');
metadata = metadata.addAttributes('Units', 'Kelvin', 'Model', 'CESM 1.0');
20  % Initialize a new, empty gridfile to catalogue the dataset
modelOutput = gridfile.new('Temperature-CESM', metadata, 'overwrite');
23  % Catalogue the source files for the dataset in the gridfile
dimensionOrder = ["lon","lat","time"];
output1Metadata = metadata.index('time',     1:12000);
output2Metadata = metadata.index('time', 12001:13872);
modelOutput.add('netcdf', outputFile1, "TREFHT", dimensionOrder, output1Metadata);
modelOutput.add('netcdf', outputFile2, "TREFHT", dimensionOrder, output2Metadata);
31  % Convert loaded data from Kelvin to Celsius
modelOutput.transform('plus', -273.15);
35  %% gridfile: Organize climate proxy records
37  % File holding the proxy record dataset
proxyFile = 'ntrend.mat';
40  % Define metadata for the proxy record dataset
info = load(proxyFile, 'years', 'site_IDs', 'lons', 'lats', 'seasons');
site = [info.site_IDs, info.lats, info.lons, info.seasons];
```




```
proxyMetadata = gridMetadata('site', site, 'time', info.years);

45   % Catalogue the proxy record dataset in a gridfile

proxies = gridfile.new('ntrend', proxyMetadata, 'overwrite');

proxies.add('mat', proxyFile, 'crn', ["time" "site"], proxyMetadata);

49   % Indicate that -999 is a fill value and should be converted to NaN

proxies.fillValue(-999);

53   %% State vector: Design and build a state vector ensemble

55   % Initialize a state vector and label the object

sv = stateVector('NTREND Assimilation');

58   % Initialize variables that are:

59   % 1. Used to run the proxy forward models, or

60   % 2. Reconstruction targets.

proxyNames = proxyMetadata.site(:,1);

sv = sv.add(proxyNames, modelOutput);

sv = sv.add(["T", "T_index"], modelOutput);

65   % Specify that ensemble members will be selected from each year of model output

66   % (Note that we are only using January as a reference month for each year.

67   % Later steps will specify the monthly means used in the assimilation).

january = month(metadata.time) == 1;

sv = sv.design(-1, 'time', 'ensemble', january);

71   % Design the reconstruction targets to use data north of 35N and a seasonal

72   % mean over June, July, and August. Also use a latitude-weighted spatial

73   % mean for the reconstructed extratropical temperature index.

extratropical = metadata.lat > 35;

JJA = [5 6 7];

latWeights = cosd(metadata.lat(extratropical));

sv = sv.design(["T","T_index"], 'lat', [], extratropical);

sv = sv.mean(  ["T","T_index"], 'time', JJA);

sv = sv.weightedMean("T_index", ["lat" "lon"], {latWeights, []});

82   % Design the proxy variables to use the site-specific seasonal temperature

83   % mean from the model grid closest to the proxy site.

nProxies = numel(proxyNames);

proxyCoordinates = str2double( proxyMetadata.site(:,2:3) );

[~, latIndices, lonIndices] = dash.closest.latlon(proxyCoordinates, metadata.lat, metadata.lon);
```





```
for p = 1:nProxies
sv = sv.design(p, ["lat","lon"], [], {latIndices(p), lonIndices(p)});
season = strsplit(proxyMetadata.site(p,4), ',');
season = str2double(season) - 1;
sv = sv.mean(p, 'time', season);
end
95   % Build a state vector ensemble with 1000 members and save it to file
[ens, ensMeta] = sv.build(1156, 'sequential', true, 'file', 'ntrend-ensemble', 'overwrite', true);
99   %% PSM: Implement forward models for the NTREND sites and estimate proxy values
101  % Load coefficients of linear forward models
coeffs = load('ntrend-forward-model-coefficients');
slopes = coeffs.slopes;
intercepts = coeffs.intercepts;
106  % Design a univariate linear forward models for each proxy record
forwardModels = cell(nProxies, 1);
for p = 1:nProxies
forwardModels{p} = PSM.linear(slopes(p), intercepts(p));
111      % Indicate the row in the state vector ensemble that holds the seasonal
112      % temperature means needed to run the forward model.
row = ensMeta.find( proxyNames(p) );
forwardModels{p} = forwardModels{p}.rows(row);
end
117  % Compute proxy estimates by running the forward models on the ensemble
Ye = PSM.estimate(forwardModels, ens);
121  %% Kalman Filter: Implement a Kalman Filter and generate a reconstruction
123  % Initialize a Kalman Filter
kf = kalmanFilter('NTREND Assimilation');
126  % Select the prior. This will be the variables in the ensemble that
127  % correspond to reconstruction targets.
ens = ens.useVariables(["T","T_index"]);
ensMeta = ens.metadata;
kf = kf.prior(ens);
```



```matlab
132  % Specify the proxy records and estimates used in assimilation
Y = proxies.load;
kf = kf.observations(Y);
kf = kf.estimates(Ye);
137  % Also specify proxy error variances
R = load('ntrend-error-variances').R;
kf = kf.uncertainties(R);
141  % Implement covariance localization with a radius of 20000 km
coordinates = ensMeta.latlon;
radius = 20000; % km
[wloc, yloc] = dash.localize.gc2d(coordinates, proxyCoordinates, radius);
kf = kf.localize(wloc, yloc);
147  % Also calculate a latitude-weighted, spatial mean temperature index from
148  % the updated spatial field. This will return the full posterior ensemble
149  % for the index, without needing to save the (very large) posterior
150  % ensemble of the full spatial temperature field.
rows = ensMeta.find('T');
latWeights = cosd(coordinates(rows,1));
kf = kf.index('T_index2', 'mean', 'rows', rows, 'weights', latWeights);
155  % Save posterior percentiles and variance, rather than the full posterior.
kf = kf.percentiles(5:95);
kf = kf.variance(true);
159  % Run the Kalman Filter.
output = kf.run;
162  % Extract the updated spatial field and its variance. Regrid to match the
163  % dimensions of the initial climate model output
[T, Tmeta] = ensMeta.regrid("T", output.Amean, 'order', ["lat","lon"]);
Tvar = ensMeta.regrid("T", output.Avar, 'order', ["lat","lon"]);
167  % Also extract the spatial-mean time series and percentiles
index1 = output.Amean(end,:);
index1p = output.Aperc(end,:,:);
index2 = output.index_T_index2;
172  % Save for post-processing and visualization
time = info.years;
save('ntrend-reconstruction.mat','T','Tvar','Tmeta','index1','index1p','index2','time');
```



```

177   %% Optimal sensor

179   % Initialize an optimal sensor for the NTREND assimilation
os = optimalSensor('NTREND sensors');

181

182   % Use the summer-temperature spatial mean index as the sensor metric.
T = ens.useVariables('T_index').load;
os = os.metric(T);

186   % Also provide the proxy estimates and error variances for the analysis
os = os.estimates(Ye);
os = os.uncertainties(R);

190   % Run the sensor to evaluate the ability of each record to reduce uncertainty
proxyPower = os.evaluate;

193   % Save for analysis and visualization
save('ntrend-optimal-sensor', 'proxyPower', 'proxyMetadata');
```



**Example 2: Global Sea Level Pressures from the Last Glacial Maximum to Present**

```matlab
1  %% Example 2: LGM Assimilation
2
3  % Reset the random number generator so all examples are reproducible
4  rng('default');
5
6  %% gridfile: Organize climate model output
7
8  % Climate model output
slpFile  =  'PSL_cam_50yrDecMon_iCESM_LGMtoPresent.nc';
sstFile  = 'TEMP_pop_50yrDecMon_iCESM_LGMtoPresent.nc';
d18OFile = 'R18O_pop_50yrDecMon_iCESM_LGMtoPresent.nc';
13 % Define metadata for the rectilinear, atmospheric model (CAM) output
lat  = ncread(slpFile, 'lat');
lon  = ncread(slpFile, 'lon');
time = ncread(slpFile, 'time');
run  = ncread(slpFile, 'run');
camMetadata = gridMetadata('lat',lat,'lon',lon,'time',time,'run',run);
20 % Define metadata for the tripolar, ocean model (POP) output
lat = ncread(sstFile, 'TLAT');
lon = ncread(sstFile, 'TLONG');
site = [lat(:), lon(:)];
popMetadata = gridMetadata('site',site,'time',time,'run',run);
26 % Initialize a gridfile object for each variable
slp  = gridfile.new( 'SLP', camMetadata, 'overwrite');
sst  = gridfile.new( 'SST', popMetadata, 'overwrite');
d18O = gridfile.new('d18O', popMetadata, 'overwrite');
31 % Catalogue the output files
slp.add('netcdf',  slpFile,  "PSL", [ "lon", "lat","time","run"], camMetadata);
sst.add('netcdf',  sstFile, "TEMP", ["site","site","time","run"], popMetadata);
d18O.add('netcdf', d18OFile, "R18O", ["site","site","time","run"], popMetadata);
37 %% gridfile: Organize climate proxy records
39 % File holding the proxy record dataset and metadata
proxyFile = 'proxies.mat';
42 % Define metadata for the proxy record dataset
```



```matlab
info = load(proxyFile, 'ID', 'lat', 'lon', 'species', 'type', 'time', 'timeUnits');
site = [info.ID, info.lat, info.lon, info.type, info.species];
proxyMetadata = gridMetadata('site', site, 'time', info.time);
47   % Catalogue the proxy record dataset in a gridfile
proxies = gridfile.new('proxies', proxyMetadata, 'overwrite');
proxies.add('mat', proxyFile, "Y", ["time","site"], proxyMetadata);
52   %% Design and build state vector ensemble
54   % Initialize and label a stateVector object
sv = stateVector('LGM Assimilation');
57   % Initialize variables
58   %    Reconstruction target:  Sea level pressure (SLP)
59   %    Forward models inputs:  Sea surface temperature (SST),  d18O of sea water
60   sv = sv.add(["SLP","SST","d18O"], [slp;sst;d18O]);
61
62   % Specify that ensemble members should be selected from the different runs
63   % and time slices. Use January of each time slice as a reference point
64   january = camMetadata.time(:,1)==1;
65   sv = sv.design(-1, ["time","run"], 'ensemble', {january,[]});
66
67   % Use annual mean values for all variables
68   months = (0:11)';        % Indices of months relative to the January reference point
69   sv = sv.mean(-1, 'time', months);
70
71   % Build the state vector sequentially using all available 50-year bins.
72   % Save to the file "lgm-ensemble.ens"
73   [ens, ensMeta] = sv.build('all', 'sequential', true, 'file', 'lgm-ensemble', 'overwrite', true);
74
75   % There are 16 50-year bins for each of the nine 3,000 year intervals.
76   % Use the sets of 16 ensemble members to build an evolving ensemble for the
77   % nine intervals.
members = 1:ens.nMembers;
members = reshape(members, 9, 16)';
ens = ens.evolving(members);
83   %% PSM: Implement forward models for the proxy sites and estimate proxy values
85   % Download proxy forward models from Github
PSM.download('bayfox');
```





```matlab
87   PSM.download('bayspline');
88
89   % Design a forward model for each proxy record
nProxies = numel(info.ID);
forwardModels = cell(nProxies, 1);
for p = 1:nProxies
94       % Get the proxy ID, location, and type
ID          = proxyMetadata.site(p,   1);
coordinates = proxyMetadata.site(p, 2:3);
coordinates = str2double(coordinates);
type        = proxyMetadata.site(p,   4);
100      % Use either a UK'37 forward model, which requires SSTs as input
if type == "uk37"
model = PSM.bayspline;
SST  = ensMeta.closestLatLon("SST", coordinates, 'site', [1 2]);
model = model.rows(SST);
106      % Or a d18O_c model, which is calibrated to different foraminiferal species...
elseif type == "d18Oc"
species = proxyMetadata.site(p,5);
model   = PSM.bayfox(species);
111          % ...and which requires SSTs and d18O_sw as input
SST  = ensMeta.closestLatLon( "SST", coordinates, 'site', [1 2]);
d18O = ensMeta.closestLatLon("d18O", coordinates, 'site', [1 2]);
model = model.rows([SST;d18O]);
end
117      % Label and save the model for each proxy
model = model.label(ID);
forwardModels{p} = model;
end
122  % Run the forward models on the inputs from the ensemble to compute proxy
123  % estimates and proxy error variances
[Ye, R] = PSM.estimate(forwardModels, ens.load);
R = squeeze(mean(R,2));
127  %% Kalman Filter
129  % Initialize and label a kalman filter
kf = kalmanFilter('LGM Assimilation');
```



```matlab
131
132 % Provide the proxy records, prior, estimates, and error variances to the filter
Y = proxies.load;
kf = kf.prior(ens);
kf = kf.observations(Y);
kf = kf.estimates(Ye);
kf = kf.uncertainties(R);
139 % Run the Kalman filter. To conserve memory, return the mean and variance
140 % of the posterior ensemble, rather than the complete ensemble
kf = kf.variance(true);
output = kf.run;
144 % Regrid the SLP reconstruction target
[SLP, SLPmeta] = ensMeta.regrid("SLP", output.Amean, 'order', ["lat","lon"]);
SLPvar         = ensMeta.regrid("SLP", output.Avar, 'order', ["lat","lon"]);
148 % Save for visualization
save('lgm-reconstruction', 'SLP', 'SLPmeta', 'SLPvar');
152 %% Proxy validation: Run the proxy system models on the posterior ensemble
154 % Run the proxy forward models using inputs from the posterior ensemble
Ypost = PSM.estimate(forwardModels, output.Amean);
157 % Save for post-processing and visualization
save('lgm-proxy-validation', 'Ypost', 'proxyMetadata');
```