# Peer review of "DASH: A MATLAB Toolbox for Paleoclimate Data Assimilation"

_EGUsphere, 2023_

## Author Response (AR1)

Reviewer 1:

*King et al. describe a new, publicly-available toolbox for paleoclimate data assimilation. The manuscript provides a description of the key concepts underlying DA, a tutorial with direct reference to examples coded in MATLAB, and a discussion of some of the benefits and pitfalls of DA in the paleosciences.*

*This toolbox and its accompanying documentation are excellent resources for the paleoclimate community. I think that the manuscript is well deserving of publication in GMD. In particular, I appreciate its readability and accessibility, which hopefully will help reduce some of the mystique and opacity surrounding DA methods and make them more broadly usable.*

*I am suggesting major revisions given the number of comments, though I do not think that addressing them will be particularly onerous. I did not work my way through the examples, but I believe that they are adequately well documented to allow future researchers to retrace their steps. I also attended a workshop demonstrating the use of this code and found those (very similar) examples clear and well-executed.*

Thank you for this positive review! Below we detail our responses to the comments. Reviewer comments are denoted in italics, and our responses are in plain-text.

*Comments*

*Is DASH an acronym? If so, please spell it out.*

DASH was originally an acronym, but that acronym is no longer an accurate description of the toolbox. As such, DASH is simply a name now.

*l1: I would not refer to paleoclimate DA as novel; these approaches have been around for 20+ years*

Thank you, we have removed the word "novel" from line 1. We also removed "novel" from the fourth paragraph of the introduction (line 58).

*l39: I'm not sure about the context for "equilibrium" here… is it the disequilibrium of climate that is at issue, or the imprint of anthropogenic forcing?*

Here it is the disequilibrium of the climate (as caused by anthropogenic forcing) that is the issue. We have rewritten this sentence to clarify this point. (lines 39-41)

*l58: There is a substantial literature using DA approaches in ocean GCMs for paleoceanography not referenced here. Approaches using the adjoint of ocean models (4DVAR) include Winguth et al. 2013; Kurahashi-Nakamura et al. 2014; Dail and Wunsch 2014; Kurahashi-Nakamura et al. 2017, and Amrhein et al. 2018. See also the work of Jake Gebbie and Olivier Marchal using simplified transport models. An early example combining paleoceanographic data with a simple ocean model is Legrand and Wunsch (1995).*

Thank you for the suggestions! We have added these references to the text (lines 59-61). We also reference several of these in Section 5 (lines 768-769).

*l63: Given the discussion in 5.2, a caveat seems in order here that while nearly any parameter can be reconstructed, there is not necessarily a guarantee of skill in doing so*

Thank you for this excellent suggestion. We have added a paragraph to the introduction that provides an overview of some of the limitations and weaknesses of DA (lines 72-77), and we include this point in that paragraph (line 73).

*l68: DA models can relax assumptions (they do not do so by default)*

Thank you, we have added "can" to this line. (line 69)

*l112: typo, Finally,in -> Finally, in*

Thank you, we have reworded this line slightly, and have removed the "in". (line 122)

*l130: This expression describes offline approaches, but not necessarily online ones, where X_p is a function of observations at previous times*

Thank you for this feedback. To clarify this, we have moved the paragraph on online vs. offline regimes immediately after this (now lines 145-153), and we have added a line explicitly noting that – in the online mode - X_p is a function of previous observations (147-148).

*l138: I suggest ", a state vector might also..." since this is not always the convention elsewhere where a state vector represents a particular time.*

Thank you – we have changed this to "possibly" (line 156)

*l146: "time step": This is a bit confusing in reference to a proxy record; is this not another example of a time slice, just referenced to a model?*

Here we were trying to distinguish between the climate model's internal time step, and time-averaged output. We have changed this to remove the phrase "time step", and instead explicitly note that state vector data may be time averaged. (lines 157, 164)

*l166: An additional important category of errors are representativeness errors that arise from comparing proxy values at, e.g., points in space or time to model values representing time slices or large grid boxes*

Thank you. We have added representativeness errors to this discussion. (lines 188-189)

*l172: "...into the climate models" -- I suggest rephrasing since this doesn't reflect what is happening in offline DA*

Thank you for the suggestion – we have altered this line accordingly. (line 194)

*l183: "help quantify" -- in fact, they do quantify the uncertainty, at least so far as the KF is concerned*

It's true that this distribution quantifies the uncertainty according to the Kalman filter – but the Kalman filter uncertainty is a subset of the total reconstruction uncertainty. For example, the total reconstruction uncertainty might also consider the effects of the choice of reconstruction methodology (e.g. Kalman filter vs particle filter vs another method), which is not captured by the Kalman filter ensemble. As such, we feel that "help quantify" is most appropriate here.

*l185: delete second )) on R*

We have fixed this typo. (line 201)

*l190: I thought this was a nice description of particle filters.*

Thank you!

*ll216 and 218: These expressions could use some more context. What is k? What is s? Note s was recently used for particle filter weights. Often argmax has a subscript denoting which argument is being chosen to maximize an expression; that would be helpful here.*

Thank you for the feedback. We have rewritten these equations to use the previous Y and R terms, albeit with a k subscript indicating the *k*th proxy. We use this k term in the equation 10, as it indicates the selection of a particular proxy and avoids confusion with the "s" term from the particle filter. We have also added the requested subscript to the argmax expression. (lines 227-234)

*l220: For this discussion (and others, e.g. proxy error) I think it would be useful to introduce the equation y = hx + n connecting proxies and the state.*

Thank you for this advice. We have added an equation (equation 2, lines 173-176) noting that

Proxy estimates = HX

To connect the proxies to the state. We have not included the full form of the suggested equation, as it is equivalent to the equation for the innovation (n = y – hx)

*l409: For context you may wish to reference the meteorological approach of "super-observations" (e.g. Purser et al. 2020).*

Thank you for this suggestion. We have not added this reference, but this is because we believe that the newly added discussion of sequential processing (lines 391-396) helps provide the context for this statement. As such, we instead note that the use of sequential processing is the cause of error-variance requirement (429-430). We also note that we have rewritten this paragraph to help provide more clarity to this section in general. (lines 429-434)

*l411: I think that you mean that the error-covarying proxies are processed using a single block update? Also, I am unsure of the example of assimilating CFRs given here. Presumably one gets a non-diagonal R from the CFR, and then there is a SVD or similar super-obbing procedure prior to block assimilation? It could be helpful to spell this out more.*

Yes, the network of error-covarying proxies is processed using a single block update.

Regarding the second comment, we think some of the confusion here results from how the "update" command differs from the classical optimal sensor algorithm (which is implemented by the "run" command). We have clarified the text here to indicate that we are only assessing the total variance reduced by the network, and not selecting independent grid cells or svd modes within the network. (lines 429-434)

*l630: Does the off-diagonal R change how DASH works under the hood? Does it use sequential observation processing for diagonal R?*

We have added several lines to section 3.2.4 explaining that DASH always uses an "all-at-once" block update, and never sequential processing. We also note that the block-update always permits off-diagonal R, so the presence of off-diagonal R does not alter how DASH works under the hood. (lines 391-396)

*l661: Variance adjustment methods may be common, but I think it's important to note here that there is no simple fix for the variance loss issue. What happens to uncertainties when those methods are used?*

Thank you for this note. We have added a line to this effect and note that uncertainties will rise when adjusting variance. (line 684-685)

*Sec. 5.2: Consider referencing Amrhein et al. (2020), which quantifies errors from model covariance bias in paleo DA using imperfect-model experiments. These biases can also lead to large errors in estimates of uncertainty in offline EnKF.*

Thank you for this suggestion – we have added this reference to the discussion. (line 707)

*l673: distal -> distant*

Thank you, we have fixed this. (line 775)

*l674: I think that accuracy (more than underlying complexity) is the more immediate issue*

We have reframed this line around model accuracy, rather than complexity. (line 697)

*l692: assume -> assumes*

We have fixed this typo (line 715)

*l745: Again, I encourage referencing the literature on adjoint approaches in paleoceanography, which conserve physical properties in the ocean.*

Thank you for the suggestion, we now reference adjoint methods in paleooceanography in this discussion. (line 697-699)

*l746: I think this is a nice articulation of an oft-overlooked point.*

Thank you!

*l763: I disagree that ensuring homogeneity of updates (which seems very unlikely!) is necessary for computing means. However, it is important to take into consideration the uncertainty of underlying grid boxes when computing means, so that regions that are less updated by data have larger uncertainties. These errors then propagate into the estimate of the index. See, e.g., Wunsch (2006), "Discrete Inverse and State Estimation Problems" (linked*

*here: http://sites.science.oregonstate.edu/~restrepo/577/wunschbook.pdf), Section 2.7.5, for a discussion.*

Thank you for this excellent point. We have altered the discussion to recommend that users propagate grid box uncertainties into the uncertainty of the index, rather than recommend that they ensure the spatial homogeneity of the updates. (lines 788-789)

Reviewer 2:

*General comments*

*This manuscript was a pleasure to read and will be an excellent contribution to Geoscientific Model Development. King et al., present a Matlab toolbox, DASH, for data assimilation of paleoclimate data. While the concepts are not new, the implementation and presentation of a well-documented toolbox is novel and very welcome. Data assimilation is becoming increasingly relevant and important to the paleoclimate community. The DASH toolbox will greatly enable the use of DA by scientists who are interested in the techniques but might have a large barrier to entry due to the coding and data management required. As a proxy-paleoclimatologist who does not currently use DA but has been intrigued by it, I found this paper and the accompanying resources available through MATLAB to be clear, helpful, and well-executed. The documentation of the code and the demos are pleasantly thorough.*

Thank you for this review! Below we detail our responses to the comments. Reviewer comments are denoted in italics, and our responses are in plain-text.

*My only minor suggestion is that the introduction somewhat glosses over limitations to DA apart from the technical barriers to implementation. The paragraph from line 57-70 is a great description of the strengths of DA, but a naive reader might think that DA is a spotless approach to paleoclimate reconstructions. The limitations/common mistakes in DA are treated thoroughly in the discussion at the end. I wonder if it would be useful to mention these limitations/pitfalls in the introduction, and to point out that a well-documented toolbox like DASH might help a new user avoid pitfalls.*

Thank you for this excellent suggestion. We have added a new paragraph that provides an overview of the limitations and caveats discussed in Section 5. We then refer readers to that section for a more detailed discussion. (lines 72-77)

*Minor Technical Corrections:*

*Line 86-87: define LMR and PHYDA*

Thank you for catching this – we now define these names. (lines 93-94)

*All figures: consider labeling the panels A, B, C, etc., for easy reference in the caption and text*

Thank you. We have added these labels and have altered the text accordingly.

*Figure 3: missing axes labels (temperature, years)*

Thank you for catching this. We have added in the missing labels.